# A Global Forest Burn Severity Dataset from Landsat Imagery (2003–2016)

Kang He[1,2], Xinyi Shen[3], and Emmanouil N. Anagnostou[1,2]

[1]Department of Civil and Environmental Engineering, University of Connecticut, Storrs, CT 06269, USA
[2]Eversource Energy Center, University of Connecticut, Storrs, CT 06269, USA
[3]School of Freshwater Sciences, University of Wisconsin, Milwaukee, Milwaukee, WI 53204, USA

*Correspondence to*: Emmanouil N. Anagnostou (emmanouil.anagnostou@uconn.edu)

**Abstract:** Forest fires, while destructive and dangerous, are important to the functioning and renewal of ecosystems. Over the past two decades, large-scale, severe forest fires have become more frequent globally, and the risk is expected to increase as fire weather and drought conditions intensify. To improve quantification of the intensity and extent of forest fire damage, we have developed a 30-meter resolution Global Forest Burn Severity (GFBS) dataset of the degree of biomass consumed by fires from 2003 to 2016. To develop this dataset, we used the Global Fire Atlas product to determine when and where forest fires occurred during that period and then we overlaid the available Landsat surface reflectance products to obtain pre-fire and post-fire normalized burn ratios (NBRs) for each burned pixel, designating the difference between them as dNBR and the relative difference as RdNBR. We compared the GFBS dataset against the Canada Landsat Burned Severity (CanLaBS) product, showing better agreement than the existing MODIS-based global burn severity dataset (MOSEV) in representing the distribution of forest burn severity over Canada. Using the in situ burn severity category data available for the 2013 wildfires in southeastern Australia, we demonstrated that GFBS could provide burn severity estimation with clearer differentiation between the high-severity and moderate/low severity classes, while such differentiation among the in situ burn severity classes are not captured in the MOSEV product. Using the CONUS-wide Composite Burn Index (CBI) as a ground truth, we showed that dNBR from GFBS was more strongly correlated with CBI ($r = 0.63$) than dNBR from MOSEV ($r = 0.28$). RdNBR from GFBS also exhibited better agreement with CBI ($r = 0.56$) than RdNBR from MOSEV ($r = 0.20$). On a global scale, while the dNBR and RdNBR spatial patterns extracted by GFBS are similar to those of MOSEV, MOSEV tends to provide higher burn severity levels than GFBS. We attribute this difference to variations in reflectance values and the different spatial resolutions of the two satellites. The GFBS dataset provides a more precise and reliable assessment of burn severity than existing available datasets. These enhancements are crucial for understanding the ecological impacts of forest fires and for informing management and recovery efforts in affected regions worldwide.

## 1. Introduction

In recent years, many regions around the world have experienced an increase in the frequency, intensity, and extent of wildfires (Doerr and Santín, 2016; Shukla et al., 2019; Dupuy et al., 2020). Wildfires are now among the most popular research topics as a result of this rising global concern, which is further heightened by changes expected in fire regimes as a consequence of changes in climate and land use (Moreira et al., 2020). While most wildfires occur in grasslands and savannas (Scholes and Archer, 1997; Abreu et al., 2017), forest fires are more dangerous and destructive and perhaps of greater interest because of their importance to the functioning and renewal of ecosystems (Flannigan et al., 2000; Nasi et al., 2002; Flannigan et al., 2006). Changes brought by the warming climate, which has dried fuels and lengthened fire seasons across the globe (Jolly et al., 2015), are also particularly significant to forested ecosystems with abundant fuels (Kasischke and Turetsky, 2006; Aragão et al., 2018).

With the rapid development of remote sensing techniques, more frequent observations from satellites facilitate the monitoring of global fire activities. The valuable information they provide at fine spatial and temporal resolutions can be used to study the number and size distributions of individual fires (Archibald and Roy, 2009; Hantson et al., 2015; Oom et al., 2016), fire shapes (Nogueira et al., 2016; Laurent et al., 2018), and locations of ignition points (Benali et al., 2016; Fusco et al., 2016). Among the most widely accepted techniques are those based on the Moderate Resolution Imaging Spectrometer (MODIS) (Chuvieco et al., 2016), which retrieves information on the entire Earth in 36 spectral bands every one to two days. The MODIS-derived burn area (BA) products are essential for ascertaining the patterns of fire occurrence, extent, propagation (Rodrigues and Febrer, 2018), and frequency (Andela et al., 2019). Based on these products, an essential indicator called "burn severity" has been derived for determining the degree of biomass consumption and the overall impact of fire on ecosystems (Keeley, 2009).

Traditionally, burn severity could be quantified from satellite sensors through spectrum information. The changes caused by fire to near-infrared (NIR) and shortwave infrared (SWIR) reflectance are highly sensitive to, respectively, canopy density and moisture content (Chuvieco, 2010). Several burn severity datasets have been generated and released based on this method. Regionally, the Monitoring Trends in Burn Severity (MTBS) dataset, which includes burn severity assessments for the contiguous United States (CONUS) and provides information on fire perimeters and severity classes, uses satellite data—specifically, Landsat imagery (Eidenshink et al., 2007). Similarly, the Canadian Landsat Burn Severity (CanLaBS) product uses Landsat imagery to assess, and map burn severity at a national scale (Guindon et al., 2021). Globally, MOdis burn SEVerity (MOSEV) has provided monthly burn severity data with global coverage at 500m spatial resolution, based on MODIS Terra and Aqua satellites (Alonso-González and Fernández-García, 2021). However, a dataset for assessing and mapping global forest burn severity based on Landsat at high spatial resolution (30m resolution) is not yet available. Such a product would support advances in fire management strategies and ecosystem conservation efforts, leading to more resilient and sustainable landscapes.

In this paper we describe a new global dataset comprising information on burn severity derived at high spatial resolution (30 meter) from Landsat imagery from the period 2003–2016. This dataset represents a step forward in quantifying and analyzing wildfire impact on forest ecosystems worldwide. We begin with a section detailing the input data and the algorithm used to process the dataset, as well as the analytical techniques employed. Section 3

presents the characteristics of the dataset and its performance in representing the distribution of forest fires. In the
results section, we analyze the advantages and disadvantages of the dataset and set forth its main contributions to
forest fire management strategies worldwide. The last section summarizes the primary findings and suggests possible
implications of the dataset.
**2. Data and Method**
Below we delineate the specifics of data input and pre-processing and the analytical techniques we employed to create
the dataset. The Global Fire Atlas was the main source of global fire records, which was overlaid with annual land
cover types from MCD12Q1 to determine when and where forest fires occurred. We then utilized the reflectance
information from Landsat's satellite archives to calculate burn severity indices for the burned forest areas. Finally, we
compared GFBS with the CanLaBS dataset available over Canada, and used the field assessed burn severity category
data in southeastern Australia and the CONUS-wide Composite Burn Index (CBI) as the ground truth to evaluate the
performances of GFBS relative to that of the existing MODIS-based global burn severity dataset (MOSEV).
**2.1. Input data**
The input data we used to build the GFBS dataset comprised the fire records available in the Global Fire Atlas for the
years 2003–2016 and all Landsat images for the same period.
The Global Fire Atlas tracks the daily dynamics of individual fires globally to determine the time and location
of ignition, area burned, and duration, as well as daily expansion, fireline length, velocity, and direction of spread. A
detailed description of its underlying methodology is provided by Andela et al. (2019).
The Terra and Aqua combined Moderate Resolution Imaging Spectroradiometer (MODIS) Land Cover Type
(MCD12Q1) Version 6.1 data product provides global land cover types at yearly intervals (Friedl and Sulla-Menashe,
2022). With its global coverage and the insights, it offers into the planet's diversity of land cover types, the MCD12Q1
dataset is pivotal to various ecological and environmental studies.
*Landsat 5,7,8* scene is a 16-day composite image with 7, 8, 11 surface reflectance bands. With its 30 meter
resolution and global coverage, it provides a high-quality, atmospherically corrected snapshot of the Earth's surface.
Use of the best available observations gathered over the 16-day period ensures the image is as clear and accurate as
possible, minimizing issues, such as cloud cover, that can obscure the satellite's view.
(https://developers.google.com/earth-engine/datasets/catalog/landsat ).
**2.2. Pre-processing**
To pre-process the data, we first imported individual fire polygons from the Global Fire Atlas into the Google Earth
Engine (GEE) and then collected the most recent Landsat images based on the tags demarcating the start and end times
of each individual fire. We applied a cloud- and snow-masking algorithm to remove any snow, clouds, and their
shadows from all imagery based on each sensor's pixel quality assessment band. By mosaicing the masked images,
we created a composite with the smallest possible cloud and shadow extent (https://developers.google.com/earth-
engine/guides/landsat ).

## 2.3. Algorithm overview

In the first step, we determined the forest fire polygons using the Global Fire Atlas data associated with the MCD12Q1 land cover data and then utilized reflectance information from Landsat's satellite archives to obtain the forest fire NBRs from the Landsat composites. Healthy plants absorb most of the visible light (for photosynthesis) while reflecting a large portion of the near-infrared (NIR) light. In contrast, areas that have been burned exhibit low NIR reflectance and high shortwave-infrared (SWIR) reflectance [Key and Benson, 2003; Montero et al., 2023]. This change in spectral properties is due to the loss of vegetation and the exposure of the underlying soil and charred material, which have different reflective characteristics. By computing this ratio for images taken before and after a fire, it's possible to determine the extent and severity of the burn [Cocke et al., 2005; Alcaras et al., 2022].

In the second step, we used the pre- and post-fire dates by the Global Fire Atlas data to obtain the corresponding pre- and post-fire NBRs, which allowed us to create the burn severity indices—that is, dNBR and RdNBR—based on the respective differences between them.

We took additional steps to validate the performance of the dataset by comparing the burn severity category data over southeastern Australia and CBIs over CONUS with those based on the MOSEV dataset. These steps are detailed in Sections 2.3.1, 2.3.2, and 2.3.3.

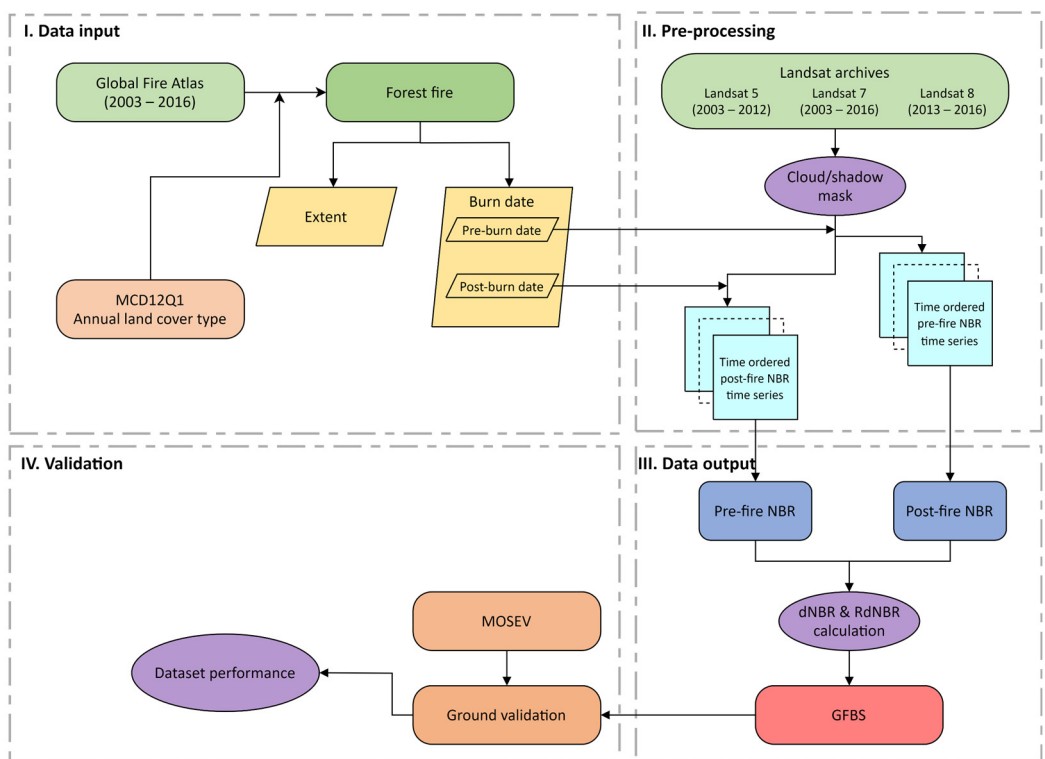

**Figure 1. Methodology for building the GFBS database (2003–2016) and validation and comparison with the MOSEV benchmark.**

### 2.3.1. Identification of global forest fires

To identify global forest fires, we first overlaid the fire polygons from the Global Fire Atlas with MCD12Q1 data from the corresponding year. Based on annual International Geosphere-Biosphere Programme (IGBP) classifications of land cover, we identified a forest fire polygon within each area where we found forest to be the dominant land cover type within the fire extent—that is, wherever the proportion of burned pixels representing forest, including evergreen needleleaf forests, evergreen broadleaf forests, deciduous needleleaf forests, deciduous broadleaf forests, and mixed forests, was largest relative to the proportion of burned pixels for other land cover types, such as shrublands and grasslands.

### 2.3.2. Estimation of the normalized burn ratio (NBR)

We calculated the normalized burn ratio (NBR) spectral index for each Landsat composite. according to the formula in Equation 1 (https://www.usgs.gov/landsat-missions/landsat-normalized-burn-ratio):

$$NBR = (NIR - SWIR) / (NIR + SWIR) \tag{1}$$

In Landsat series 4 through 7, we collected NIR information from Band 4 and SWIR information from Band 7. In Landsat 8, we collected NIR information from Band 5 and SWIR information from Band 7.

### 2.3.3. Estimation of dNBR and RdNBR

Having obtained burn area locations and burn dates from the Fire Atlas product, we selected from the Landsat 16-day time series valid pre-fire and post-fire NBR pixels that were, respectively, from the date most closely preceding the start date and the date most closely following the end date of each burned polygon within a three-month time window.

The dNBR index, calculated according to Key and Benson (2006) as shown in equation (2), is the reference burn severity spectral index used by the European Forest Fire Information System (https://effis.jrc.ec.europa.eu/about-effis) and by the United States' Monitoring Trends in Burn Severity program (https://www.mtbs.gov). Larger dNBR values indicate higher burn severity:

$$dNBR = preNBR - postNBR \tag{2}$$

RdNBR is another burn severity spectral index that is widely used, including by the United States' Monitoring Trends in Burn Severity program (https://www.mtbs.gov/, last access:1 May 2021). The RdNBR normalizes the dNBR to the square root of pre-fire NBR value, which helps in reducing the variability caused by pre-fire vegetation conditions and enhances the accuracy in assessing burn severity [Miller et al., 2009]. As formulated in equation (3) (Miller and Thode, 2007), higher RdNBR values indicate higher burn severity:

$$RdNBR = dNBR/\sqrt{|preNBR|} \tag{3}$$

**2.4. Validation**

To validate the GFBS database, we used the 112 ground-verified burn severity category data following the Fire Extent and Severity Mapping (FESM) scheme for the 2013 wildfires over southeastern Australia. The FESM severity classes include unburnt, low severity (burnt understory, unburnt canopy), moderate severity (partial canopy scorch), high severity (complete canopy scorch, partial canopy consumption), and extreme severity (full canopy consumption). Besides FESM, we used the ground-measured CONUS-wide Composite Burn Index (CBI) from 2003 to 2016. CBI was developed by Key and Benson (2006) to assess the aboveground effects of fire on vegetation and soil land use types (i.e., burn severity). It is determined through direct field observations after a fire when assessors visited various sites within the burned area to evaluate the effects of the fire on different components of the ecosystem, such as the degree of charring, percentage of foliage consumed, changes in ground cover, and mortality of plants. The CBI score for each site was calculated by averaging the scores of the different components. This overall score represents the burn severity at a specific site. The index ranges continuously from 0 (unburned) to 3 (high severity). These values have been related to satellite-derived burn severity values through regression equations (https://burnseverity.cr.usgs.gov/products/cbi). In this study, we used all available CBI values over CONUS to establish relationships between CBI and the dNBR and RdNBR values of the GFBS and MOSEV datasets. We used the Pearson correlation coefficient and bias as metrics to evaluate the performance of the two datasets. Figure 2 (a) shows the locations of the 112 ground-verified burn severity sites for the 2013 wildfires over southeastern Australia. Figure 2 (b) shows the locations of CBI observations over CONUS for the period from 2003 to 2016. Of the 1,315 ground-surveyed CBI reports for forest fires during that time, most came from western states, such as Arizona, Colorado, and Oregon, where forest fires are more frequent and severe. Fewer CBI records are available in eastern states, such as Florida and Georgia.

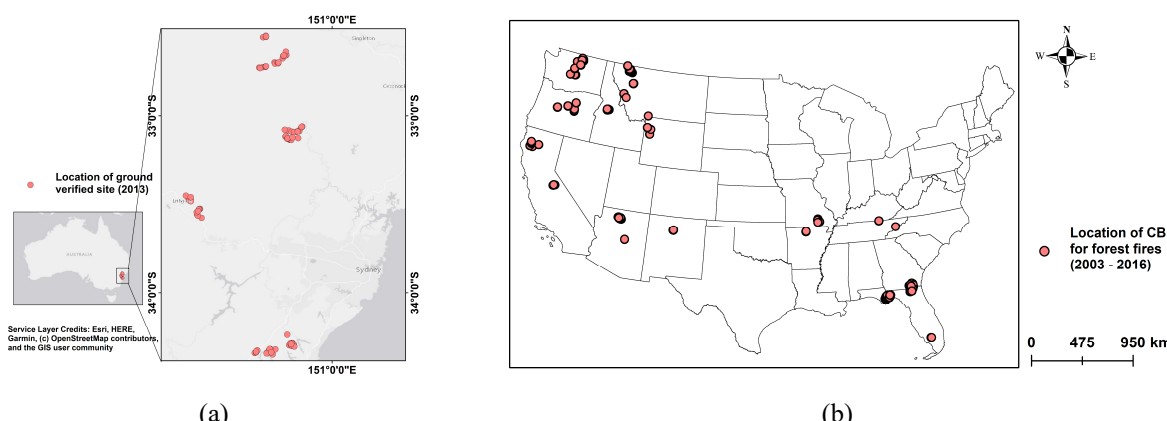

(a)                                                                                      (b)

**Figure 2. Locations of (a) ground verification burn severity sites over southeastern Australia and (b) forest fire CBIs over CONUS.**

168        In addition to validation against in-situ data., we also compared the fire severity magnitudes of GFBS with

the CanLaBS dataset available over Canada. CanLaBS provides burn severity information for burned areas identified
from the Canada Landsat Disturbance product at the level of individual 30m resolution pixels. The dataset was derived
from Landsat imagery and uses values of pre-fire to post-fire differences in dNBRs for nearly 60 million hectares of
burned areas across Canada's forests from 1985 to 2015. [Guindon et al., 2017; Guindon et al., 2018].
**3. Results**
**3.1. Forest fire coverage of Landsat composites.**
Figure 3 (a) shows the number of forest fire polygons globally between 2003 and 2016, representing individual fire
events, from the Global Fire Atlas dataset. Approximately 80,000 forest fire events occur in the world each year on
average, where more than 90,000 happened in 2004 and more than 100,000 in 2003 and 2015, respectively. Figure 3
(a) displays the availability of Landsat imagery covering the burn area where individual forest fires happened
worldwide. From 2003 to 2012, Landsat 5 could provide images covering between 35% and 68% of the recorded
forest fire events in the Global Fire Atlas, while Landsat 7 images covered 83% to 93% of the Global Fire Atlas events.
From 2013 to 2016, Landsat 7 images covered between 90% and 98% of the fire events, while Landsat 8 images
covered more than 97%. The Landsat composites combining all available Landsat 5 and Landsat 7 images from 2003
to 2012 and Landsat 7 and Landsat 8 images from 2013 to 2016 significantly increased the number of forest fires
shown by Landsat images, with coverage of the fire events ranging from 88% to 99%. Figure 3 (b) shows the
distribution of the spatial coverage of cloud-free Landsat composites for individual fires from the Fire Atlas. We used
a cloud and shadow removal algorithm to eliminate invalid poor-quality pixels from recorded forest fires, resulting in
a line chart showing the distribution of the percentages of valid pixels to the total burn pixels in each year. Overall,
the spatial coverage was above 72%, and the coverage has been above 85% since 2013, when Landsat 8 was launched.

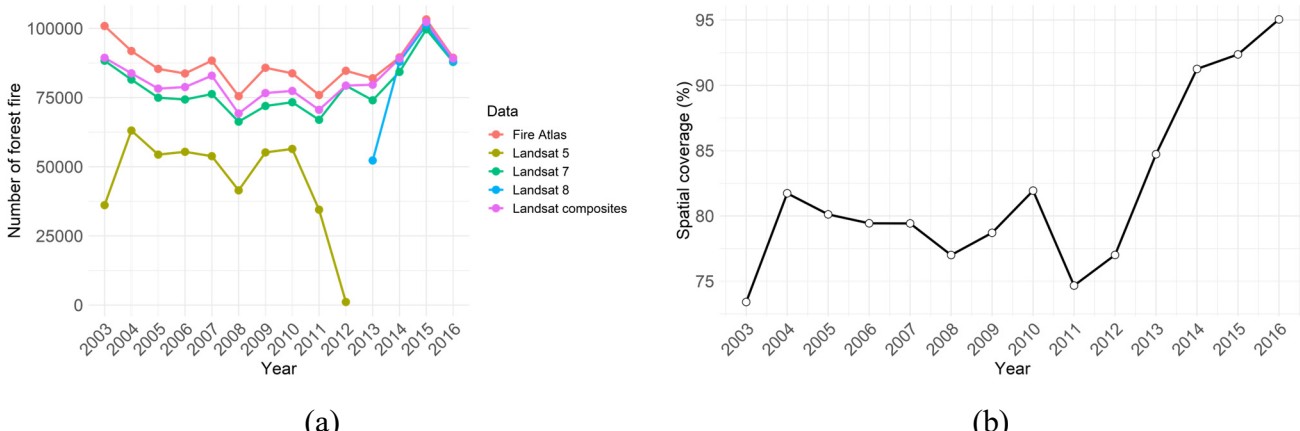

(a)                    (b)

**Figure 3. (a) Numbers of individual fires from the Fire Atlas and available Landsat imagery; (b) Spatial coverage of cloud-free Landsat composites for individual fires reported in the Fire Atlas.**


Figure 4 shows the data process for a single post-NBR Landsat composite for the fire event that ended on 17
September 2015 in north Washington. The first prior image for NBR calculation was on 20 September 2015 from
Landsat 8 (as image 1). The cloud and shadows are removed in image 1 after applying the cloud/shadow mask. The
next available image on 21 September 2015 from Landsat 7 (as image 2) was then used to fill those gaps in image 1
and obtain a new Landsat composite (phase 1). The third available image on 29 September 2015 from Landsat 8 (as
image 3), image on 15 October 2015 if needed, was adopted sequentially to fill the un-scanned gap pixels in phase 1
and generate the final post NBR image for this event. The process for pre-NBR image calculation is the same but in a
reversed time-order from the start time of the fire event.

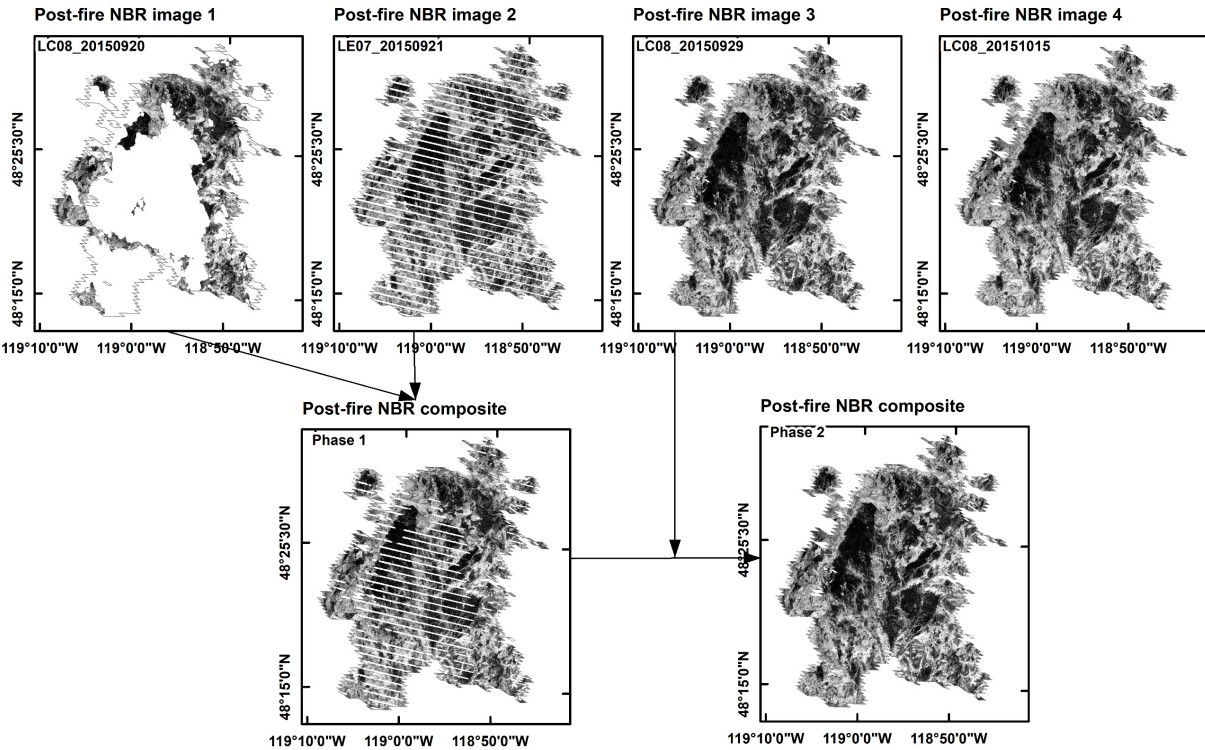

**Figure 4. NBR image process for Landsat composite, for the fire event ended on 17 September 2015 in north Washington.**


The scatterplot in Figure 5 (a) shows the NBR values of the overlapping pixels in image 1 and image 2, with
the associated distributions of NBR for the fire event. It is noted that NBR values in images 1 and 2 show high
correlation (r = 0.96), relatively low bias (-23.81%) and similar probability densities, even though they are derived
from two different Landsat images (Landsat 8 and Landsat 7). The scatterplot in Figure 5 (b) shows the NBR values
of overlapping pixels in image 1 and image 3, with the associated distribution of NBR for the fire event. Similarly,
NBR values in image 1 and image 3 have high correlation (r = 0.96) and low bias (12.30 %) and similar probability
densities, even though they are derived from different times (9 days apart). The results indicate that the cloud-free
NBR composite mosaicking of all available Landsat images has reasonable accuracy with high spatial and temporal
consistency.

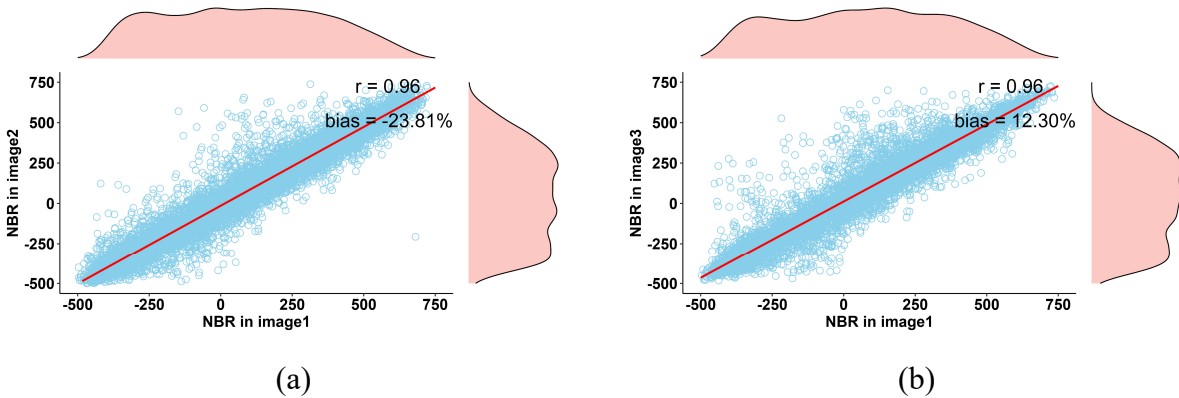

(a)                                                                  (b)

**Figure 5. Scatterplots of overlapped pixel values in (a) image 1 and image 2; (b) image 1 and image 3.**


3.2 Comparison between GFBS and CanLaBS over Canada
In this section we describe the comparison of the fire severity maps of GFBS and MOSEV datasets to the ones from
the CanLaBS dataset over Canada for an overlapped period from 2003 to 2015. Figure 6 shows the number and the
trend of forest fires over Canada from 2003 to 2015, by CanLaBS data and GFBS products, while the vertical bar
represents the number of forest fires recorded by both CanLaBS and GFBS each year. Due to the different sources
and algorithms to map the burn area, the number of forest fires depicted by CanLaBS is larger than those by GFBS
each year. Nevertheless, it is noted that GFBS agrees with CanLaBS in terms of the variations of forest fire activities,
such as the intense forest fires in 2004 and 2015 and the relatively low number of forest fires in 2007 and 2008.

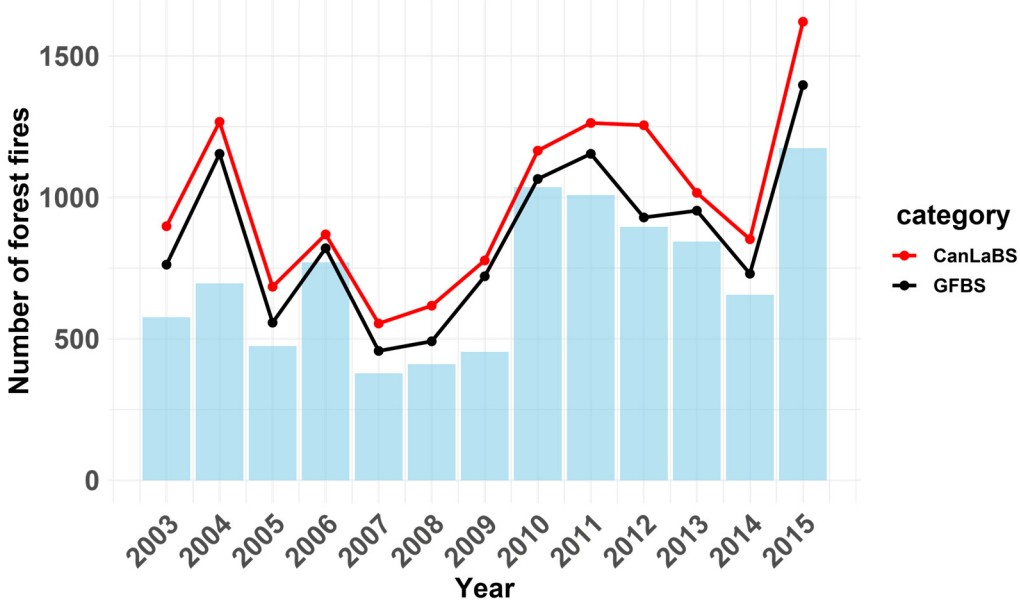

**Figure 6. Number of forest fires by CanLaBS and GFBS dataset. Vertical bars show the number of overlapping forest fires.**

217        Figure 7 illustrate the scatterplots of dNBR of forest fires from CanLaBS against those from GFBS (panel a)

and MOSEV (panel b), for the period 2003 to 2015. Consistent to the results shown in Figure 6, dNBR from GFBS
shows strong correlation with the dNBR from CanLaBS with r being 0.77 and a slightly underestimation of the overall
dNBR for forest fires (bias = -12.42%). On the other hand, dNBR from MOSEV exhibited low correlation with the
dNBR from CanLaBS  (r = 0.42) and slight overestimation (bias = 11.84 %). Figure 7 (c) displays the probability
density function (PDF) plots of CanLaBS dNBR, GFBS dNBR and MOSEV dNBR. It is noted the close PDFs of
GFBS dNBR and CanLaBS dNBR, though the mode of GFBS distribution is at slightly lower dNBR value relative to
the CanLaBS distribution. On the other hand, the distribution of MOSEV dNBR significantly deviates from CanLaBS
dNBR, having a lower peak and larger tails.

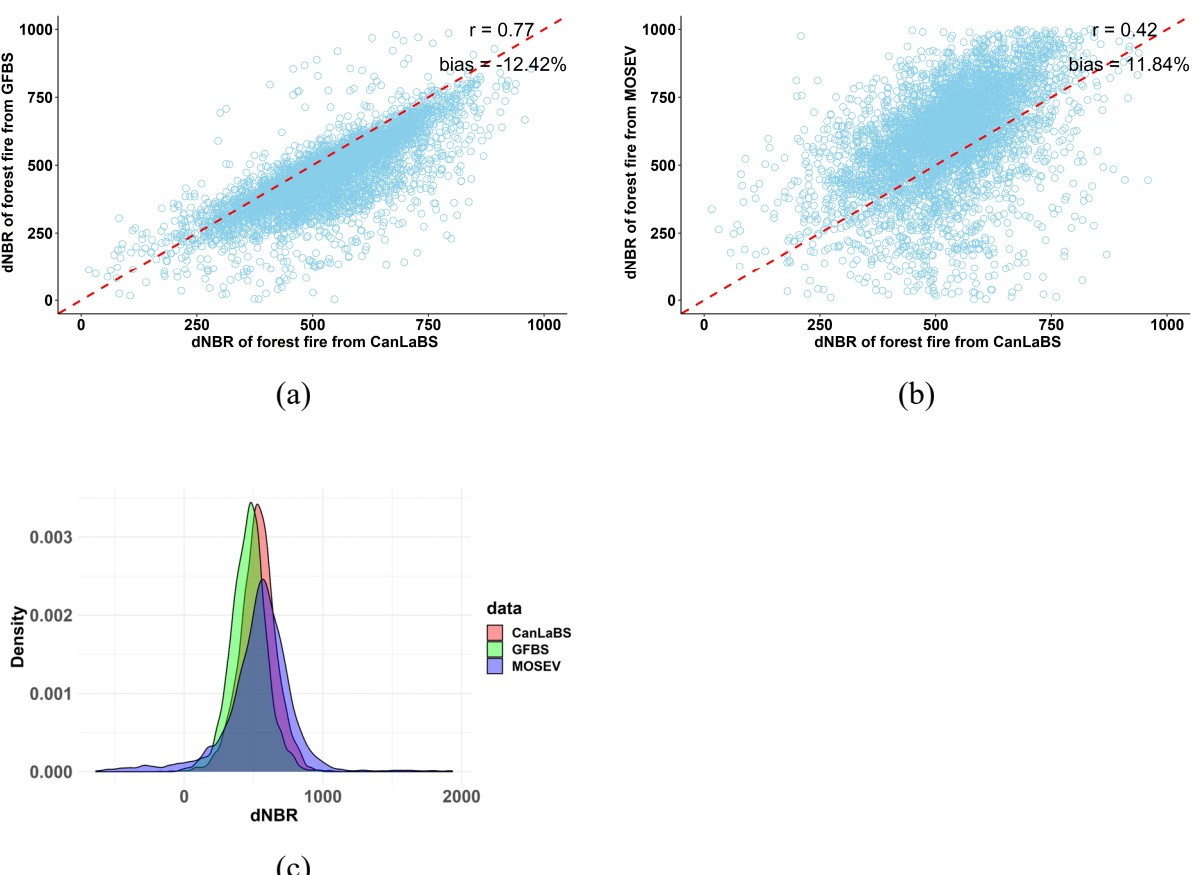

(a)

(b)

(c)

**Figure 7. Scatterplots of dNBR from CanLaBS against those from (a) GFBS and (b) MOSEV; (c) density plot of dNBR from CanLaBS, GFBS and MOSEV, for forest fires from 2003 to 2015 over Canada.**


227        Figure 8 presents the boxplots of distributions of dNBR from CanLaBS, GFBS and MOSEV separate by

year. Consistent to the previous results, GFBS compares well with CanLaBS in terms of the dNBR distribution of
annual forest fires and as well as the variations of dNBR over time, even though it provides slightly lower dNBR
values compared to CanLaBS. On the other hand, MOSEV compared poorly with CanLaBS annual dNBR
distributions, exhibiting overall larger dNBR values and larger anomalies over time.

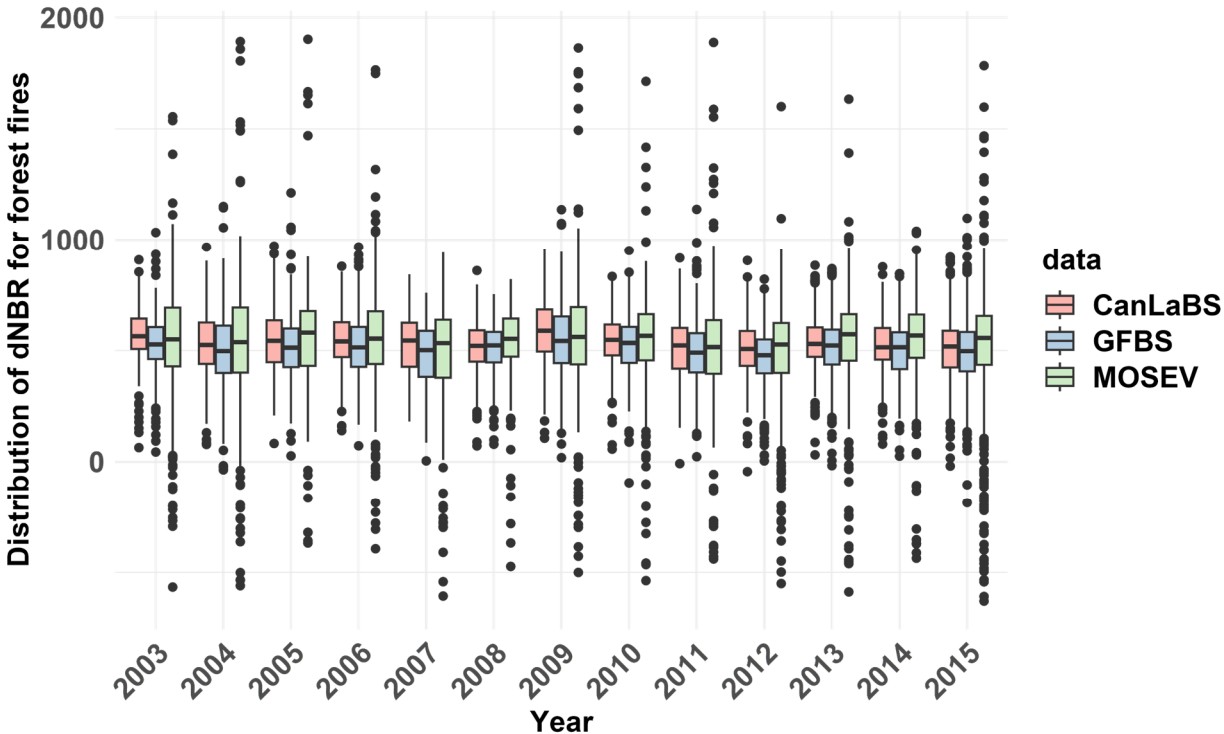

**Figure 8. Boxplots of annual distributions of dNBR values from CanLaBS, GFBS and MOSEV for forest**

**fires over Canada from 2003 to 2015.**

**3.3. Validation against in situ fire severity category over southeastern Australia**
Using as the ground truth the in-situ burn severity categorizations from the 2013 wildfires over southeastern Australia,
we evaluate the performance of GFBS and MOSEV datasets. Figure 9 (a), (b) and (c) display the spatial patterns of
GFBS dNBR and MOSEV dNBR for wildfires that happened on October 15 2023, October 17 2023 and October 21
2023, respectively, in southeastern Australia, where relatively dense in situ burn severity categorization data are
available. It is noted that GFBS dNBR shows similar spatial patterns to the MOSEV dNBR in the events on October
15 2023 and October 17 2023, both showing significant fire centers where high dNBR are found. For the October 21
2023 event, however, the dNBR map from MOSEV shows a larger high burn severity area than GFBS.

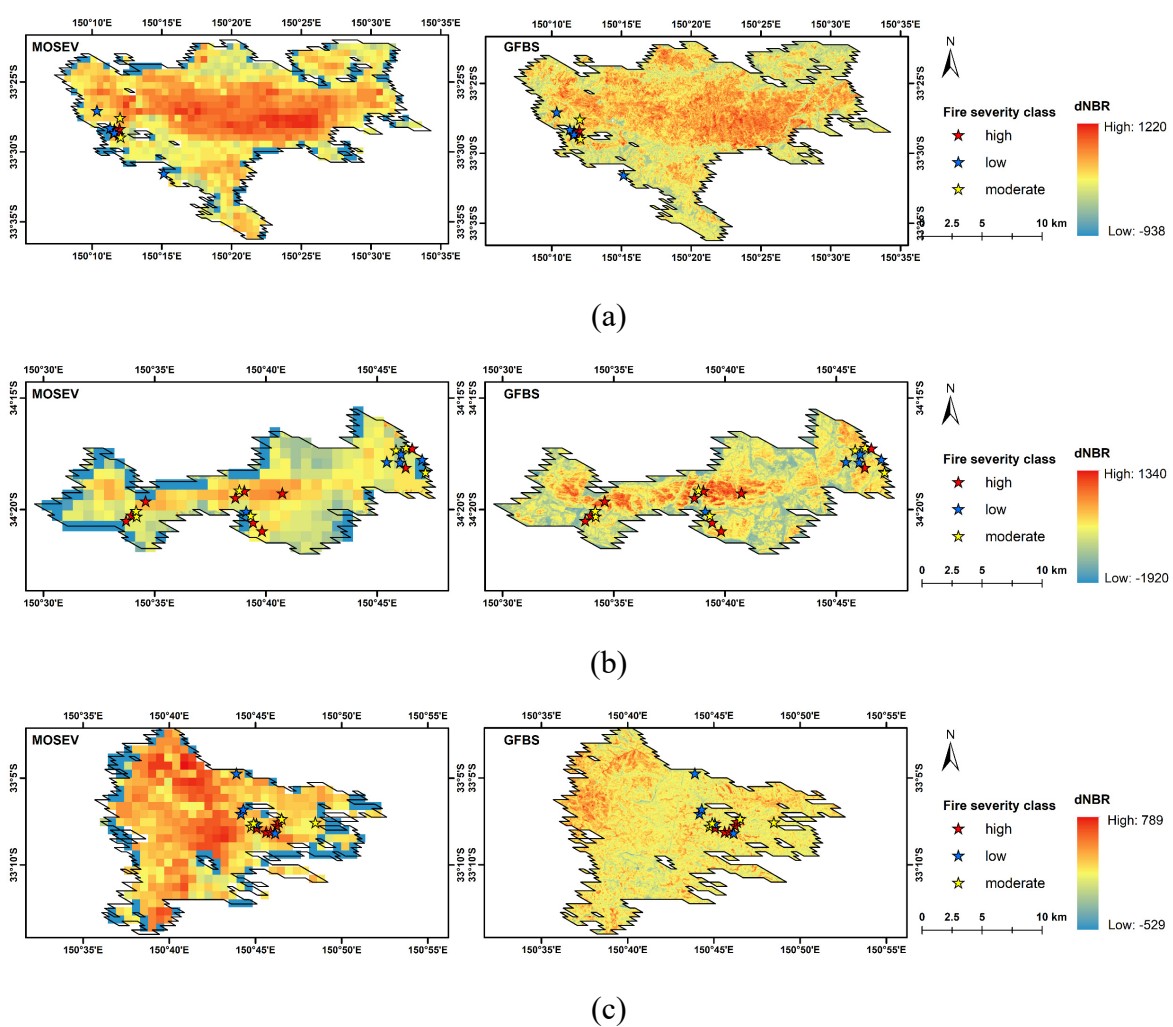

**Figure 9. Spatial patterns of dNBR for wildfires on (a) October 15 2023, (b) October 17 2023 and (c) October 21 2023, in southeastern Australia, derived from the GFBS and MOSEV datasets.**


The boxplots in Figure 10 (a), (b) and (c) display the corresponding distributions of dNBR from GFBS and MOSEV at different observed severity classes in the events on October 15 2023, October 17 2023 and October 21 2023, respectively. The severity classes, e.g. low, moderate and high, are categorized from the field assessed sites in the corresponding fire events. For the event on October 15 2023, dNBR from GFBS shows significant difference between the moderate/high and low severity class, and no difference between high and moderate severity class. The dNBR from MOSEV, however, presents lower dNBR at high severity class than those at moderate and low severity class. For the event on October 17 2023, both GFBS and MOSEV show significant discrepancies on dNBR between high and moderate/low severity class. For the event on October 21 2023, GFBS could clearly differentiate among high, moderate and low severity classes in terms of dNBR values, while MOSEV presents the lowest dNBR values at the moderate severity class, while exhibits small differences in dNBR values between the low and high severity classes. Figure 10 (d) shows the overall performances of dNBR from GFBS and MOSEV for the different severity

classes, combining all 112 ground verification sites. More significant differences are shown in the GFBS dNBR
boxplots between high, moderate and low severity classes than those from MOSEV, indicating a better skill of GFBS
to distinguish between forest fires of different severity levels.

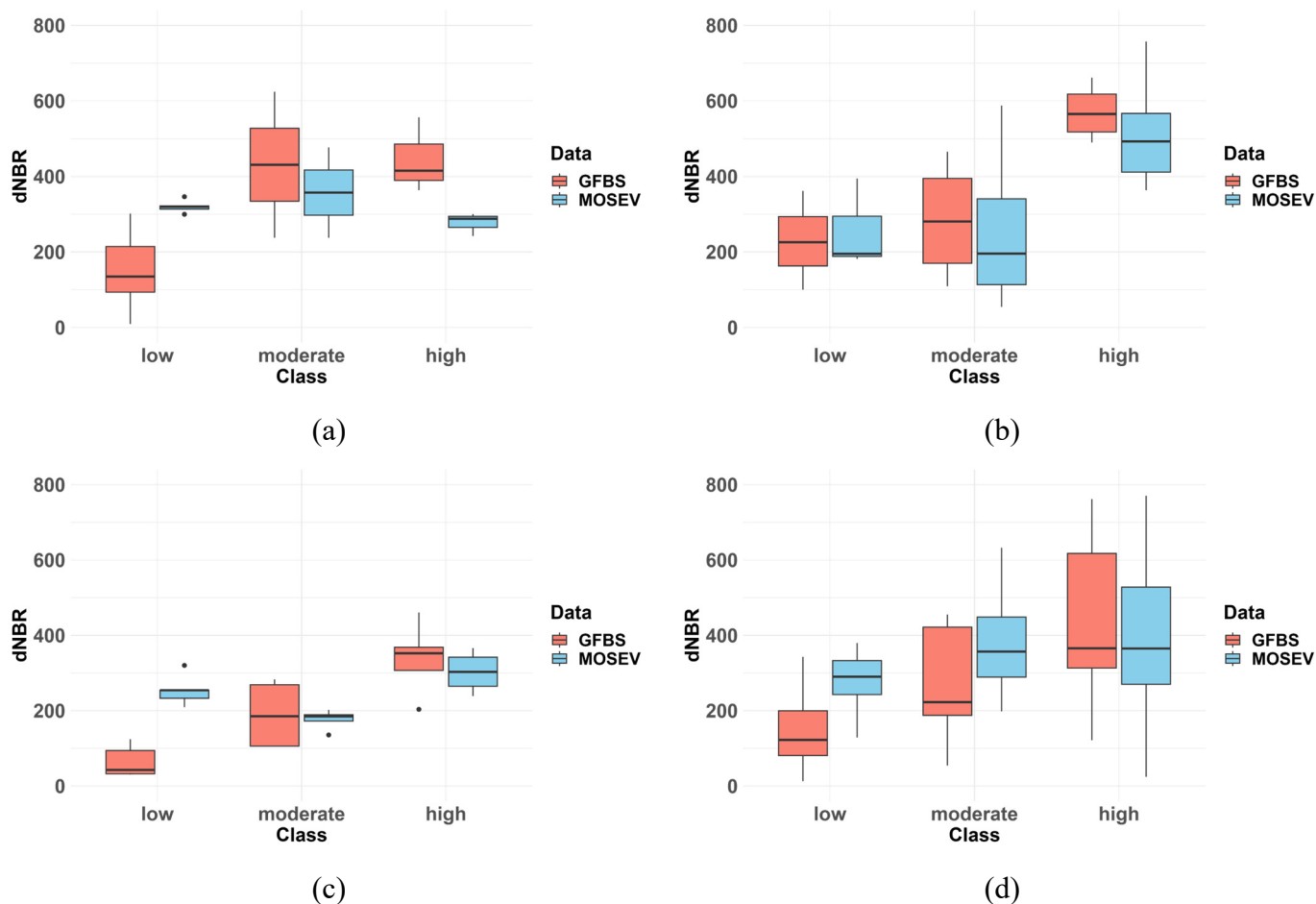

(a)                                                    (b)

(c)                                                    (d)

**Figure 10. Boxplots of distributions of dNBR at different burn severity classes from the in situ data for (a) event on October 15 2023; (b) event on October 17 2023; (c) event on October 21 2023; and (d) combining all events with in situ data.**


256          As mentioned above, MOSEV gave relatively small dNBR values in the event on October 15 2023, where

burn severity is classified from in situ measurement as high. Figure 11 (a) displays the location of the ground
verification sites with the corresponding burn severity class and associated dNBR values from MOSEV and GFBS. It
is noted that within one MOSEV grid cell (500 meter) four ground verification sites are located. The dNBR value
from MOSEV is 295 for all four sites, while three of the sites are classified as low and only one site is classified as
high severity. On the other hand, at GFBS resolution (30 meter), we can note significant spatial variation in dNBR,
with GFBS dNBR being 239 for the site classified as high and 9, 16 and 68 for the sites classified as low severity. In
a surrounding MOSEV pixel we note a site classified as high severity, but dNBR from MOSEV is 188 while dNBR
from GFBS is 397. In the event on October 21 2023, we found that MOSEV gave relatively high dNBR values at
ground verification sites that are classified as low severity. Figure 11 (b) shows the locations of ground verification

sites with corresponding classified burn severity and associated dNBR values from MOSEV and GFBS. In the two adjacent MOSEV grids, the dNBR values from MOSEV are 287 and 327 respectively where both sites are classified as low severity. At GFBS resolution more significant changes between high and low dNBR are found within the same MOSEV grid, resulting in dNBR values of 30 and 32 for the ground verification sites classified as low severity. The results demonstrate the significance of GFBS high resolution data in representing the small-scale variations of dNBR and providing more granular and reliable dNBR estimations.

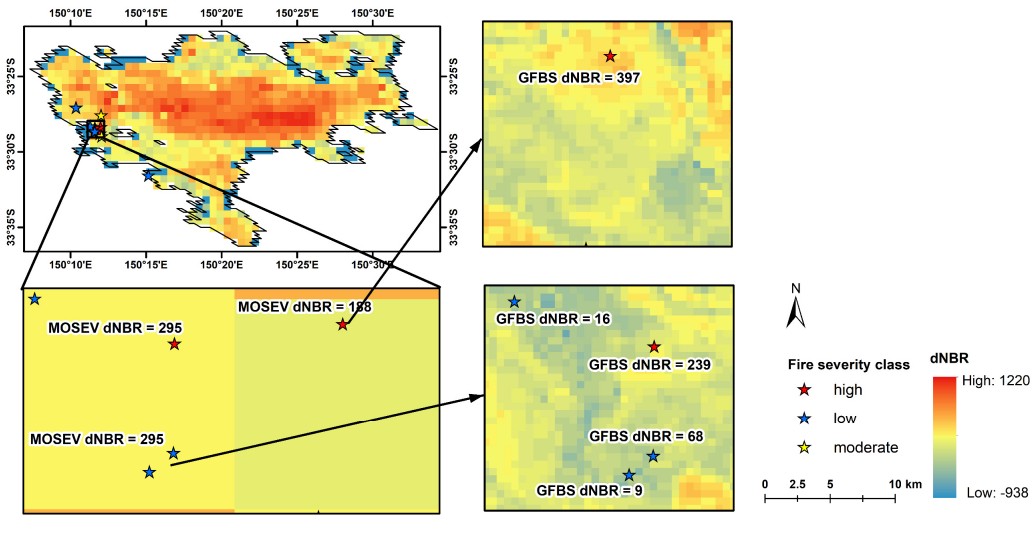

(a)

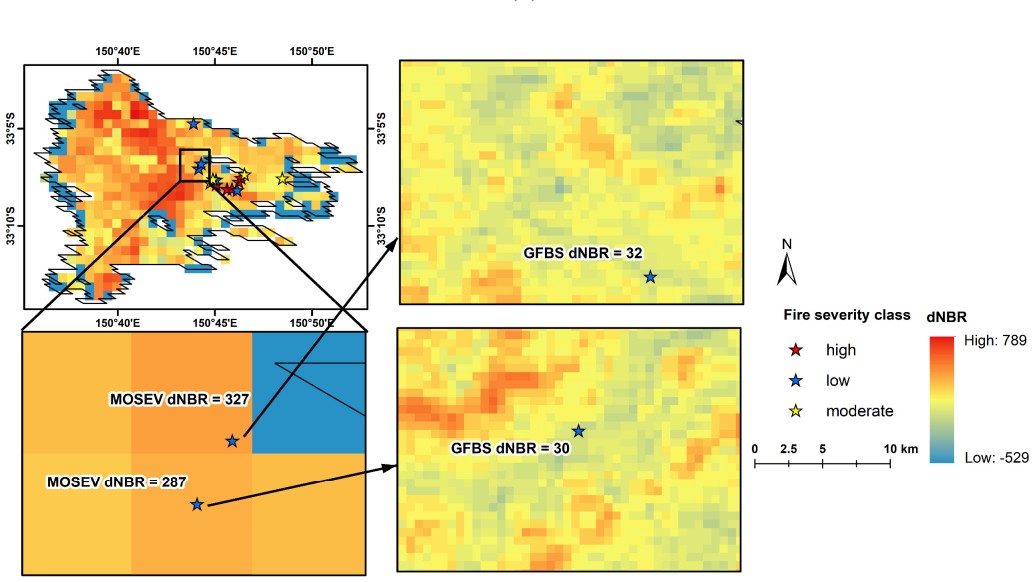

(b)

**Figure 11. The location of ground verification sites with burn severity classes overlaid by dNBR values from GFBS and MOSEV for the fire event of (a) October 15 2023 and (b) October 21 2023.**

**3.4. Validation against CBI over CONUS**

Figure 12 (a), (b), (c) and (d) shows the spatial patterns of dNBR derived from GFBS and MOSEV over CONUS for the forest fires with the largest burn areas (referred to as annual maximum forest fire hereafter) in 2004, 2006, 2007, and 2013 respectively for which CBI records are available. The figures present the PDFs of dNBR values from GFBS and MOSEV, along with spatial distribution maps of dNBR. The similarity in spatial patterns between GFBS burn severity and MOSEV burn severity is noted in these plots. Significant differences occur, however, between dNBR from GFBS and MOSEV. Specifically, MOSEV tends to provide overall larger dNBR values, but where dNBR from GFBS is relatively high MOSEV dNBR values are relatively lower. This difference could also be inferred from the PDFs of dNBR from GFBS and MOSEV where dNBR from MOSEV distributed more on the mean dNBR of around 300, while dNBR from GFBS is bimodal with peaks on both low and high values. For the annual maximum forest fire in 2007, especially, MOSEV showed more extensive areas with high dNBR values compared to GFBS, a difference that was also revealed in the large deviation of mean dNBR values in the PDFs of dNBR from the GFBS (mean dNBR around 100) and MOSEV (mean dNBR around 500) datasets.

The density plot of dNBR in Figure 12 also shows the bi-modal distribution for dNBR from GFBS, at around 100 (associated with low severity) and 700 (associated with high severity), for the annual maximum forest fire in 2006. dNBR from MOSEV on the other hand shows a single peak distribution at around 500, indicating that dNBR from MOSEV underestimated the high severity occurrences, and overestimated the low severity ones, depicted in the GFBS dataset. For the annual maximum forest fire in 2013, though the density plot presents two different peaks in the distributions of GFBS and MOSEV, indicating a significant difference in the burn severity depicted in the two datasets.

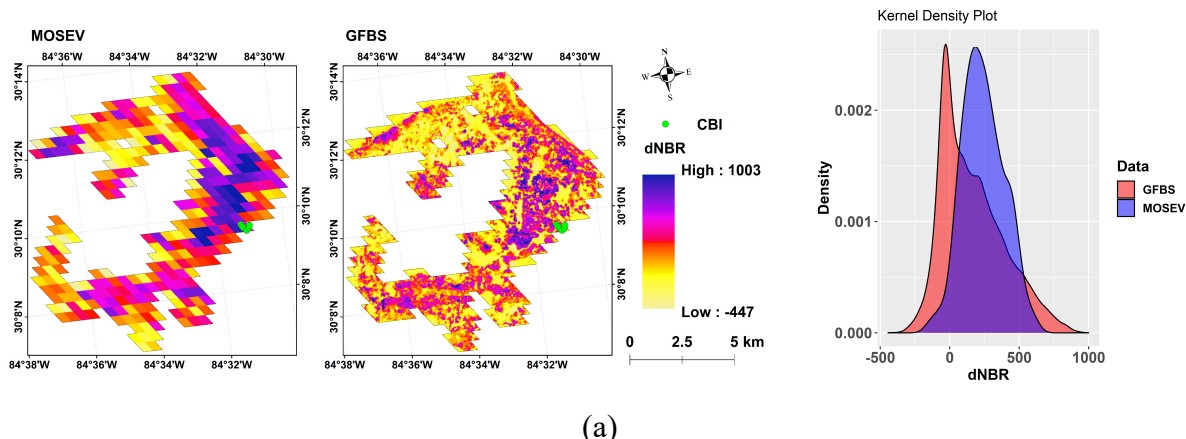

(a)

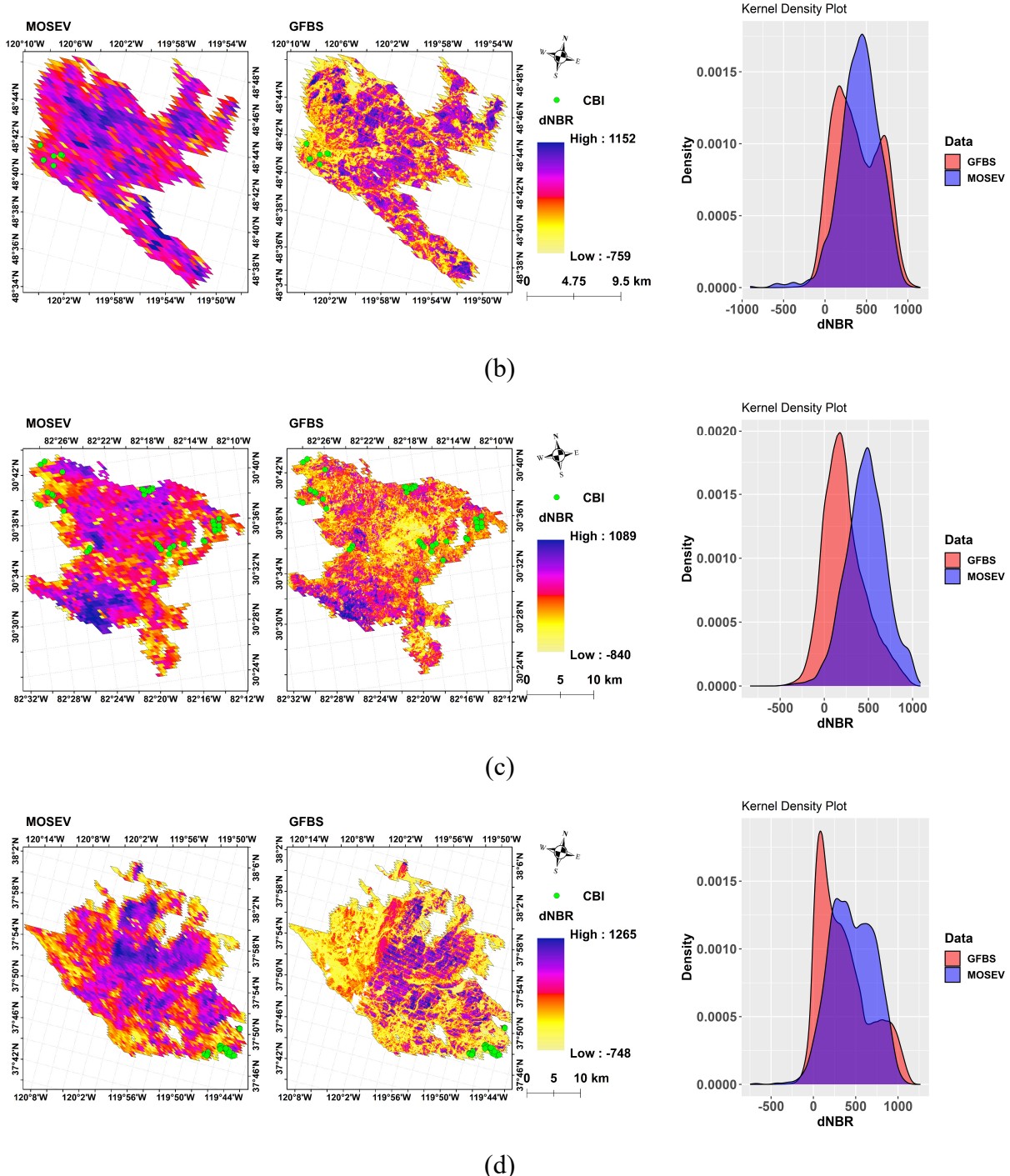

(b)

(c)

(d)

**Figure 12. Spatial patterns of dNBRs for annual maximum fires over CONUS with distribution of probability density functions in (a) 2004, (b) 2006, (c) 2007, and (d) 2013, derived from the GFBS and MOSEV datasets.**

Figure 13, panels (a), (b), (c), and (d), present the scatterplots of CBI against dNBR from GFBS and dNBR from MOSEV for the annual maximum forest fires in 2004, 2006, 2007, and 2013, respectively. For the annual maximum forest fire in 2004, Figure 13 (a) shows a positive correlation between CBI (r = 0.45) and dNBR from

GFBS, while we found no correlation between CBI and dNBR from MOSEV. For the annual maximum forest fire in
2006, we found good agreement between the CBI and dNBR from GFBS, with a r value of 0.85, while the r value was
only 0.28 for dNBR from MOSEV. Though correlations between CBI and dNBR from GFBS and MOSEV were poor,
dNBR from GFBS showed a positive trend to CBI, while the relationship between CBI and dNBR from MOSEV was
negative, for the annual maximum forest fire in 2007. For the annual maximum forest fire in 2013, dNBR from GFBS
(r = 0.72) was more strongly correlated with CBI than dNBR from MOSEV (r = 0.36).

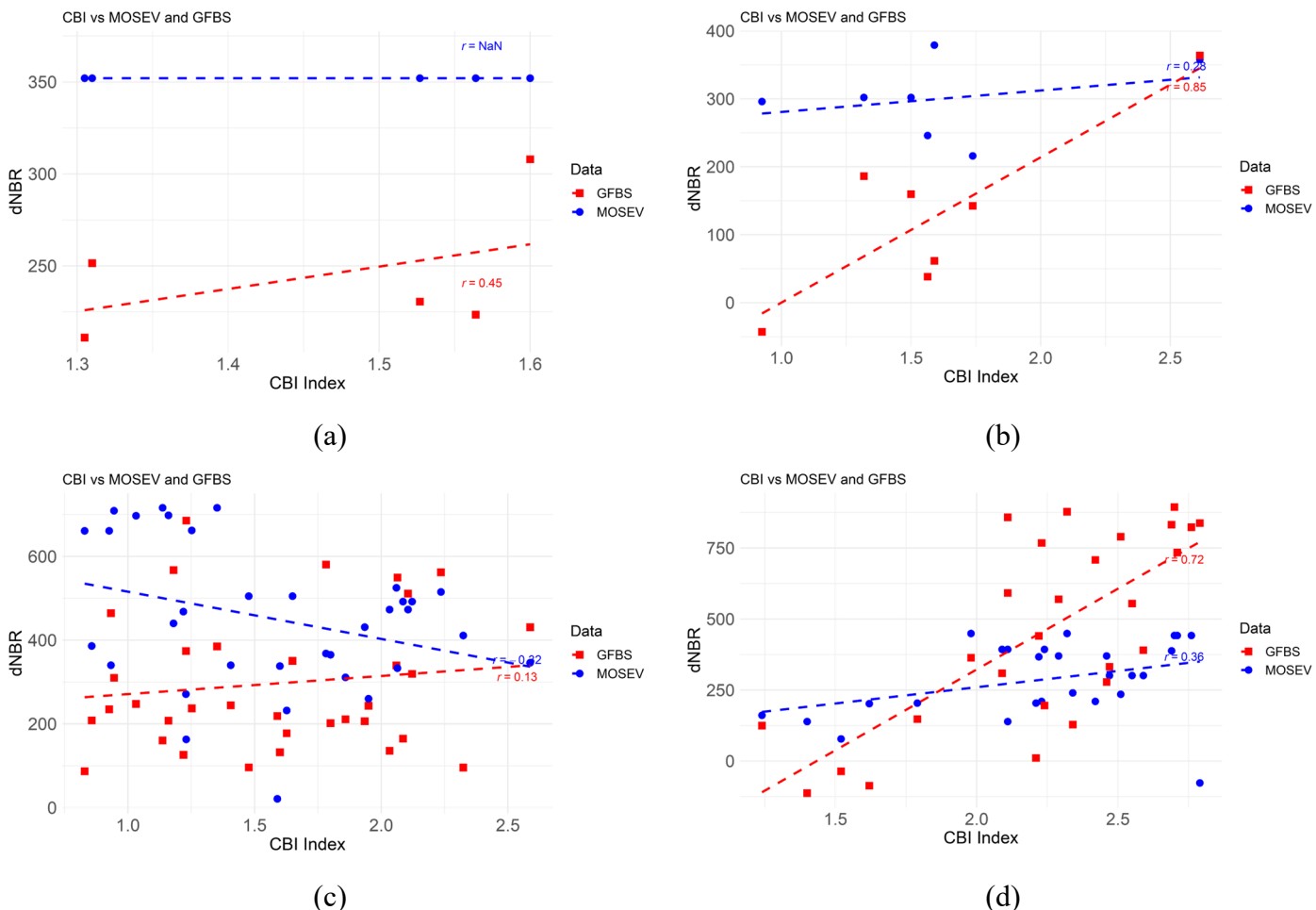

**Figure 13. Scatterplots of CBI against dNBR from GFBS and MOSEV for annual maximum fires in (a) 2004,**
**(b) 2006, (c) 2007, and (d) 2013.**
Figure 14 (a), (b), (c) and (d) shows the spatial patterns of RdNBR from GFBS and MOSEV along with the
associated PDFs of RdNBR, for the forest fires over CONUS with the largest burn areas (referred to as annual
maximum forest fire hereafter) in 2004, 2006, 2007, and 2013 respectively. RdNBR from GFBS and MOSEV exhibit
similar spatial patterns yet provide different ranges of RdNBR values over burn area. RdNBR from MOSEV tended
to be higher than RdNBR from GFBS, which is consistent to the density plots of RdNBR from GFBS. The mean value
in the distribution of RdNBRs from MOSEV is larger than the mean value in the distribution of RdNBRs from GFBS,
for the annual maximum forest fires in 2003, 2006 and 2007. The density plots of RdNBR from GFBS and MOSEV
are largely overlapped for the annual maximum forest fire in 2013, but RdNBR from MOSEV distributed more on the
mean values around 800 than RdNBR from GFBS, while RdNBR from GFBS distributed more on the extreme low
values above 0 and high values above 1500. These findings demonstrate that RdNBR from MOSEV represents overall
larger burn severity estimations than RdNBR from GFBS.

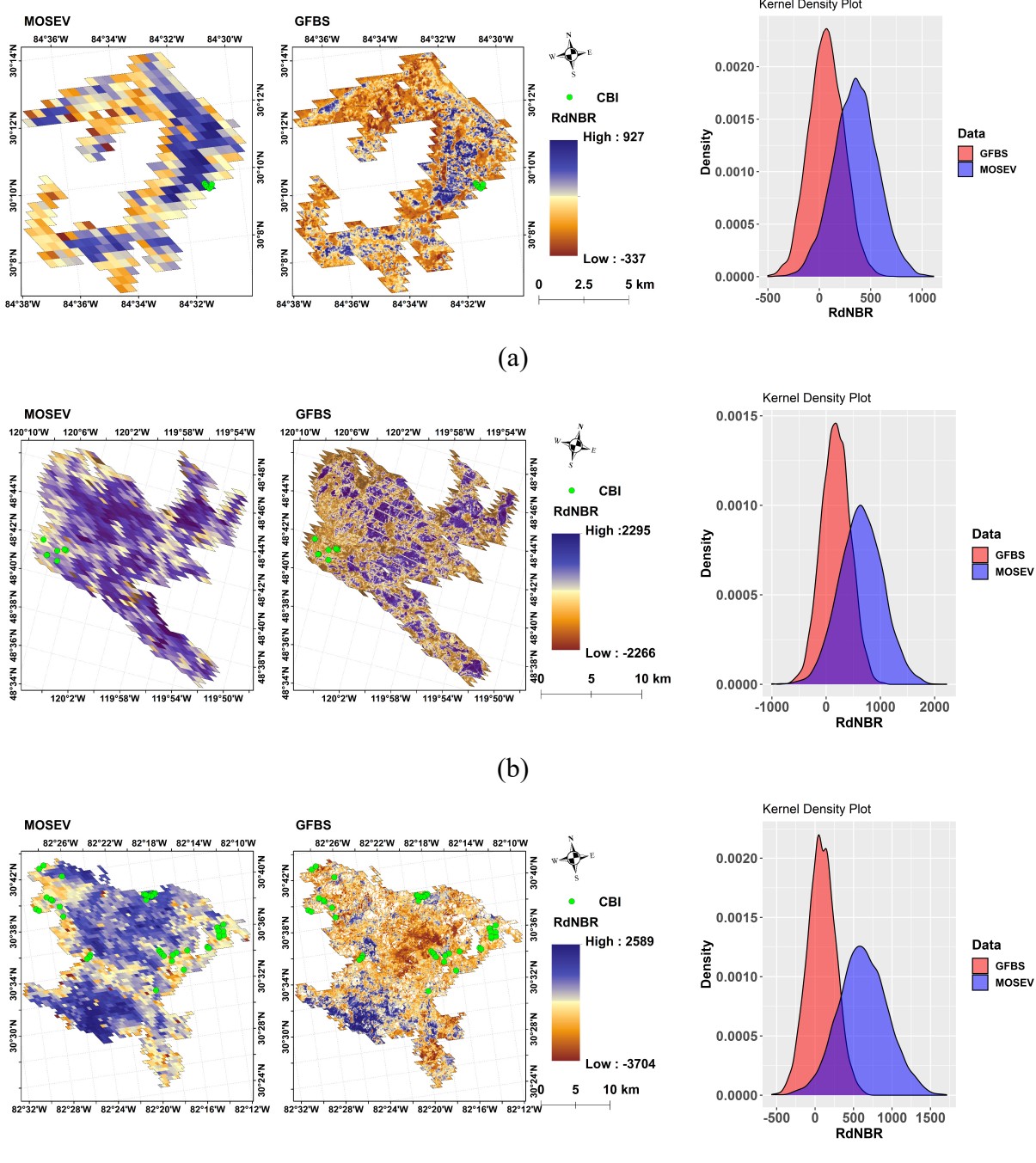

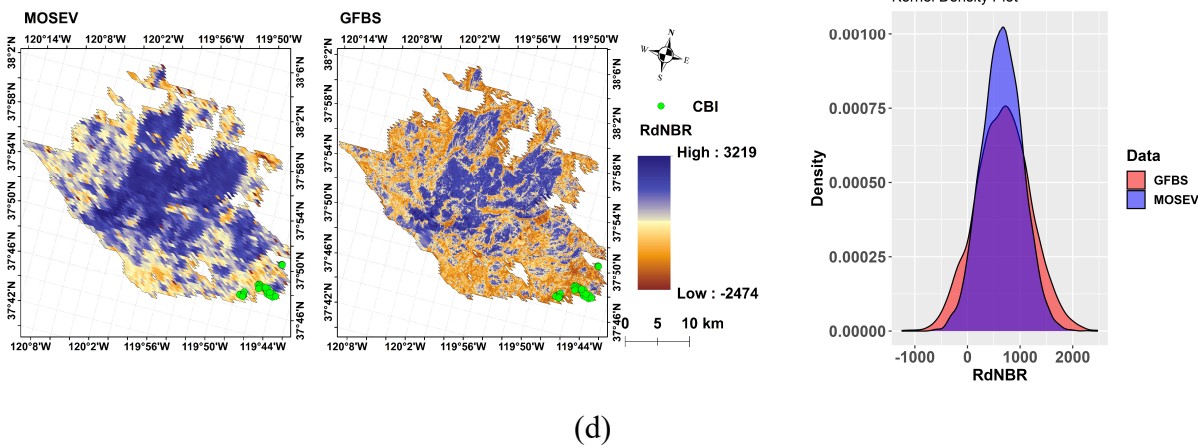

(d)

**Figure 14. Spatial patterns of RdNBRs for annual maximum fires over CONUS with distribution of probability density functions in (a) 2004, (b) 2006, (c) 2007, and (d) 2013, derived from the GFBS and MOSEV datasets.**

313        Figure 15, panels (a), (b), (c), and (d), present the scatterplots of CBI against RdNBR from GFBS and

MOSEV, for the annual maximum forest fires in 2004, 2006, 2007, and 2013, respectively. For the annual maximum
forest fire in 2004, RdNBR from GFBS shows a positive correlation with CBI (r = 0.57), while no correlation was
found between CBI and RdNBR from MOSEV. For the annual maximum forest fire in 2006, RdNBR from GFBS
correlated well with the CBI for showing a r value of 0.85, while the r value was only 0.18 between CBI and RdNBR
from MOSEV. The correlations between CBI and RdNBR from GFBS and MOSEV are bad for the annual maximum
forest fire in 2007, the RdNBR from GFBS showed a positive trend to CBI with r = 0.15, while the RdNBR from
MOSEV showed a negative trend to CBI with r = -0.28. For the annual maximum forest fire in 2013, RdNBR from
GFBS (r = 0.74) was more strongly correlated with CBI than RdNBR from MOSEV (r = 0.40).

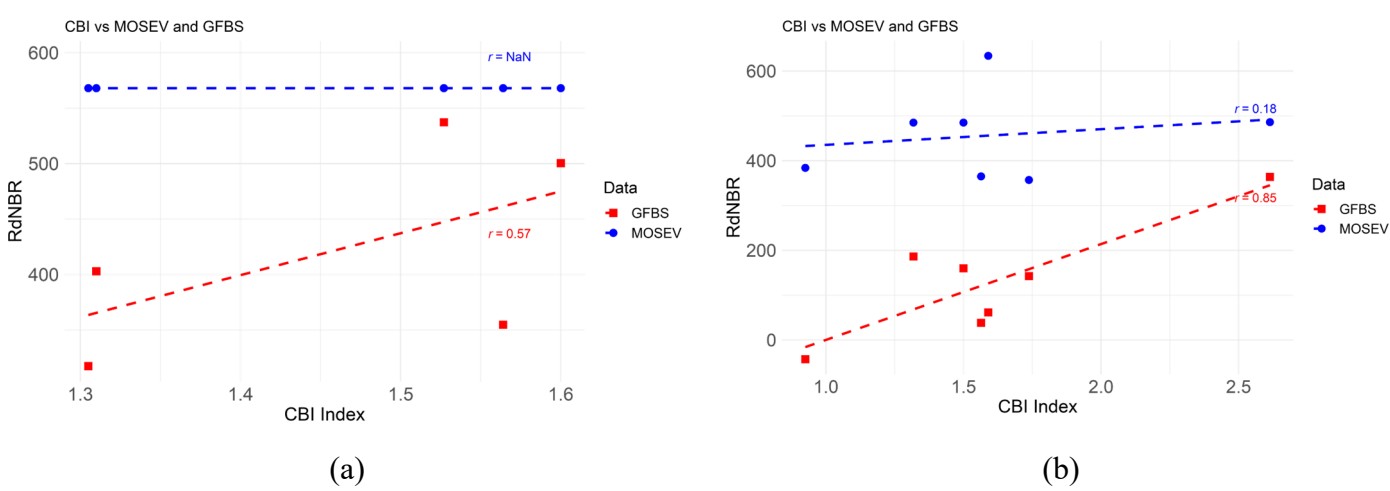

(a)                           (b)

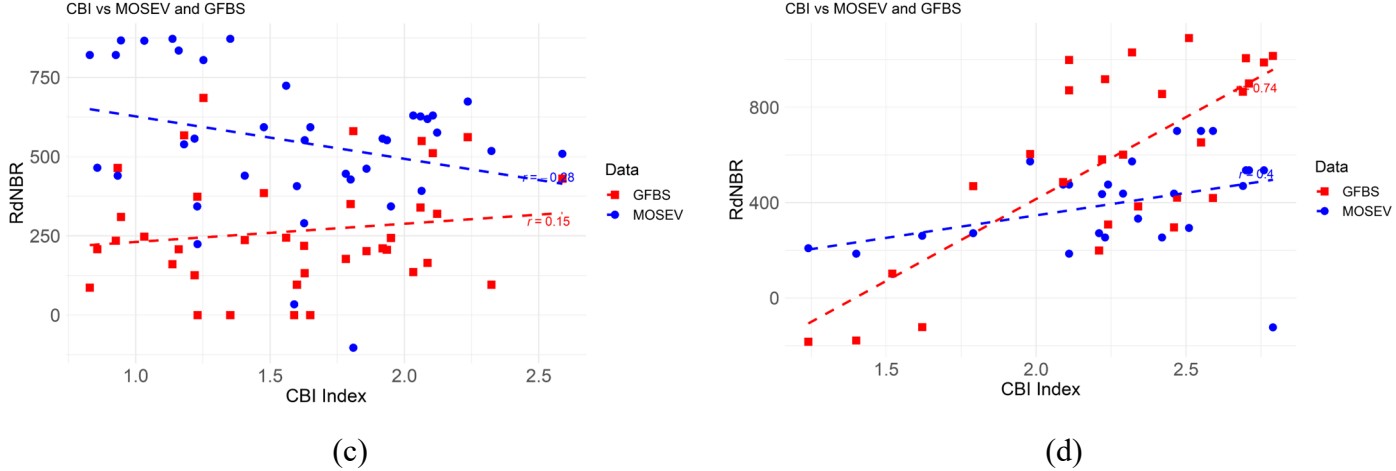

(c)                                                                                    (d)

**Figure 15. Scatterplots of CBI against RdNBR from GFBS and MOSEV for annual maximum fires in (a) 2004,**
**(b) 2006, (c) 2007, and (d) 2013.**

Figure 16 (a) and (b) shows the scatterplots of CBI against dNBR from GFBS and MOSEV, respectively, for
all forest fires from 2003 to 2016 over CONUS. Involving all ground validations, we found GFBS dNBR shows a
stronger correlation with CBI (r = 0.63) than MOSEV dNBR (r = 0.28). Using RdNBR as the burn severity, Figure
16 (c) and (d) show that GFBS RdNBR (r=0.56) outperformed MOSEV RdNBR (r=0.20).

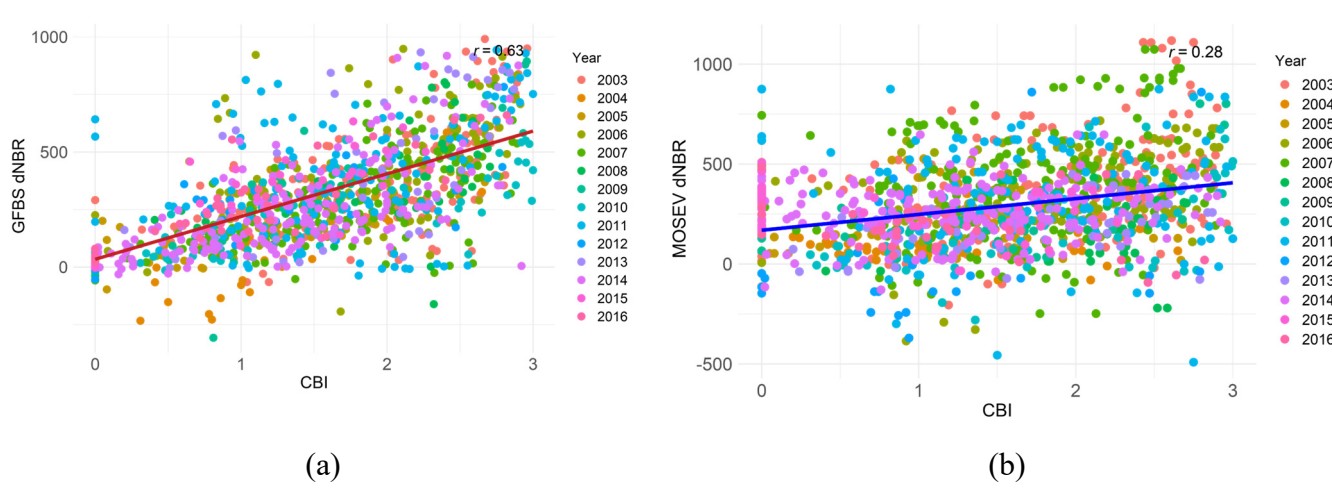

(a)                                                                                    (b)

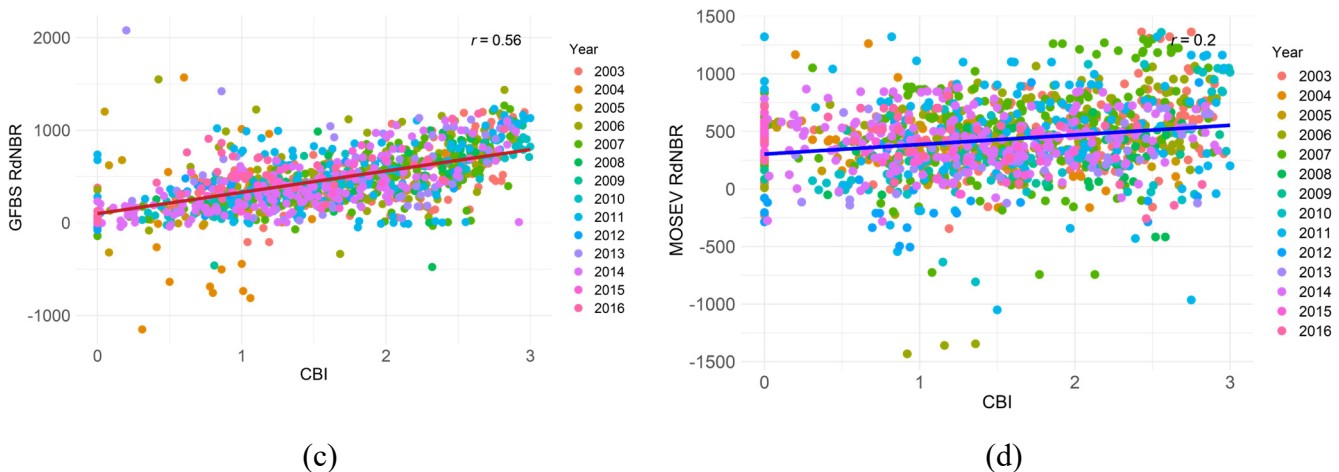

(c)                                                                    (d)

**Figure 16. Scatterplots of CBI against (a) dNBR from GFBS, (b) dNBR from MOSEV, (c) RdNBR from GFBS, and (d) RdNBR from MOSEV for forest fires from 2003 to 2016 over CONUS.**

## 3.5. Comparison of GFBS and MOSEV globally

Figure 17 (a) displays the global spatial distributions of the overlapping area between the density plots of dNBR from GFBS and MOSEV, which is defined as the area intersected by two probability density functions presented in Figure 12 and Figure 14. The overlapping areas in density plots typically represent the percentage of common values between the distributions of two datasets, which ranges from 0 to 1 with the larger value indicating the two distributions are more likely come from the same distribution. As Figure 17 (a) shows, we found the overlapping area over most of the world to be above 0.4, indicating a similarity of 40% between the burn severity information provided by GFBS and MOSEV in these regions. For some regions, like South America, Western Europe, and southeast Australia, the overlap was above 0.6.

From Figure 17 (b), which shows the global distribution of the mean dNBR for each burn pixel derived from GFBS, we found the global spatial heterogeneity of burn severity to be small, with dNBR values from GFBS around 100 and 200. The exception was in Western Europe, where dNBR was above 300. The global distribution of the mean dNBR for each burn pixel derived from MOSEV, as shown in Figure 17 (c), however, indicated a large spatial variability in burn severity globally. The MOSEV dataset, for example, indicated that the forest fires in north CONUS and Canada should have an average dNBR above 300, while in the GFBS dataset the average dNBR value was around 100 to 200. The MOSEV dataset also indicated the average dNBR values for forest fires in South Africa and China should be close to or below 0, while in the GFBS dataset they were around 100 to 200, respectively.

Figure 17 (d) presents a more detailed comparison between the dNBR from GFBS and MOSEV globally, showing the difference in the mean dNBR for each burn pixel, as calculated by dNBR from MOSEV minus dNBR from GFBS. Globally, MOSEV data indicated higher forest burn severity than GFBS over Canada and CONUS, also found in the results presented in section 3.2 and 3.4, as well as southeast Australia (also found in the results presented in section 3.3). MOSEV data presented lower forest burn severity over Mexico, South Africa, Europe, China, and

Southeast Asia. These findings revealed that the forest burn severity information provided by GFBS might be more
reliable and reasonable than that provided by MOSEV for some fire-prone areas, such as CONUS, as validated in this
study. This improved accuracy over MOSEV data would support advances in decision making in fire management
strategies and ecosystem conservation efforts.

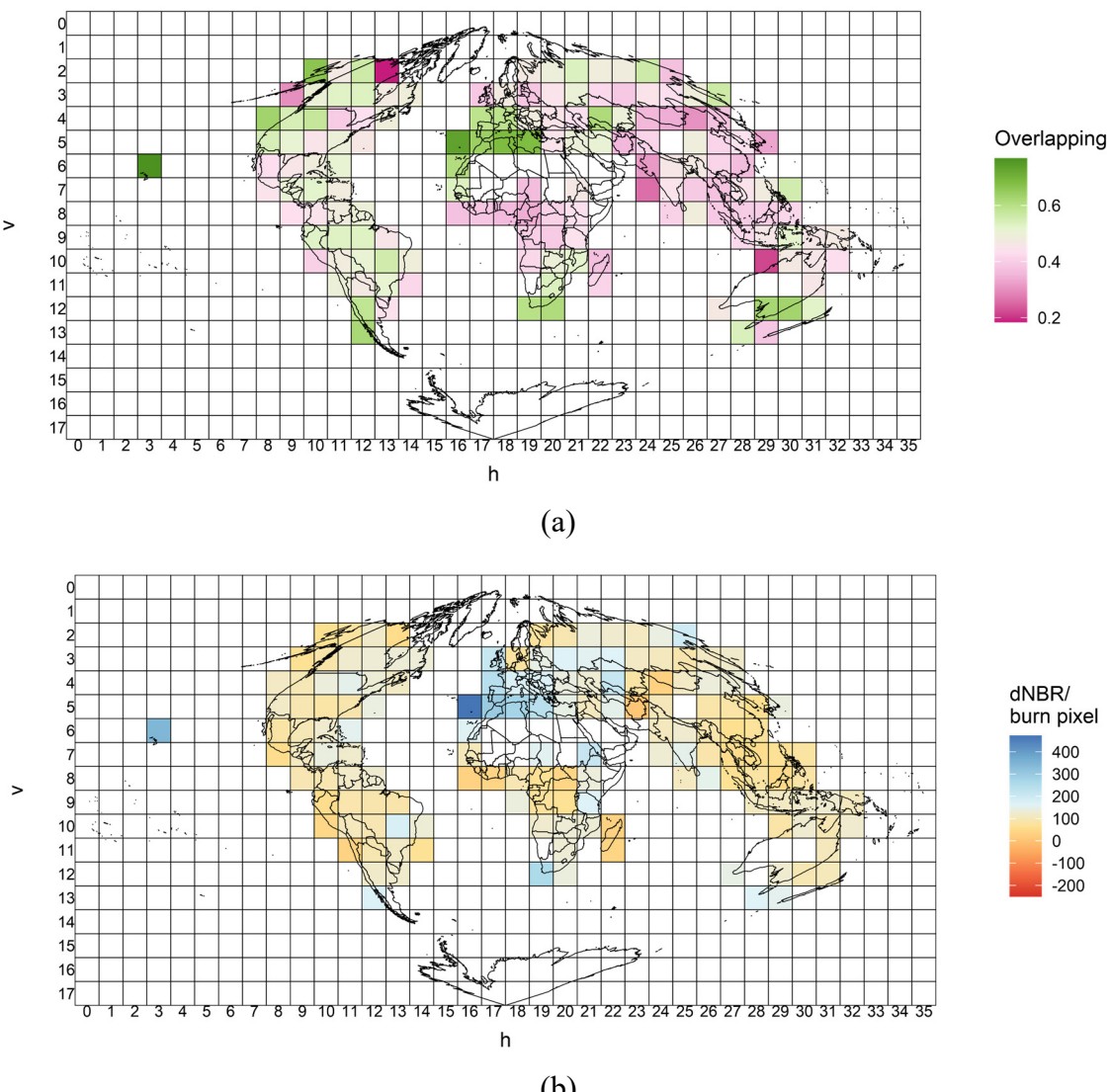

(a)

(b)

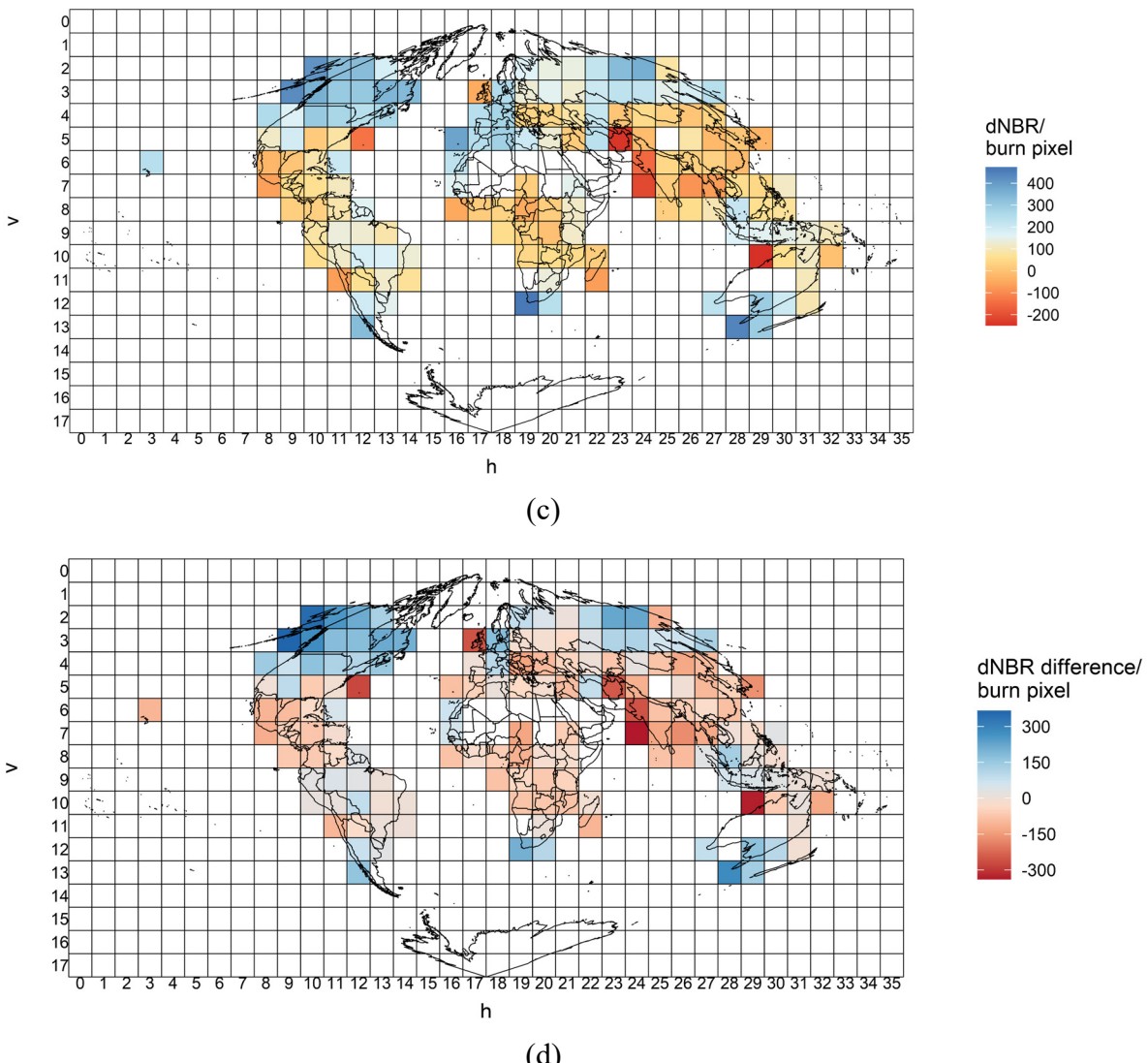

(c)

(d)

**Figure 17. Global spatial distributions of (a) overlapping areas between the density plots of dNBR from GFBS and MOSEV, (b) the mean dNBR per burn pixel from GFBS, (c) the mean dNBR per burn pixel from MOSEV, and (d) the differences in the mean dNBR per burn pixel between MOSEV and GFBS (MOSEV – GFBS).**

**4. Discussion**
The GFBS dataset presented in this paper is the first to provide fine spatial resolution (30m) burn severity information
for global forest fires from 2003 to 2016. Compared with the existing Landsat based CanLaBS dataset, GFBS shows
closer agreement to CanLaBS in describing the distribution of annual forest fire burn severity than the MODIS based
MOSEV data. As suggested by the validation against the ground reference, GFBS can better represent the spatial
variability and provide higher performance than the MOSEV dataset. In addition, GFBS is shown to have more reliable
burn severity estimations than MOSEV for some fire-prone areas, like CONUS, Canada, and Australia, which could
support advances in decision making in fire management strategies and ecosystem conservation efforts.

The difference in the performance of GFBS and MOSEV with respect to burn severity can be attributed to two sources. The first is spatial resolution. GFBS, based on Landsat (5, 7, and 8) images, is at a resolution of 30 meters, while MOSEV is based on MODIS Terra MOD09A1 and Aqua MYD09A1 images with a resolution of 500 meters. As shown in Figure 11 (a), stemming from the coarse spatial resolution, MOSEV provides dNBR value of 295 for the site classified as high severity as well as for those classified as low severity, leading to an overestimation for low severity sites. With the improved spatial resolution, GFBS is able to capture more detailed localized variability of dNBR, providing more reasonable dNBR estimation for low severity sites (dNBR equal to 9, 16, 68). Similarly, in the event shown in Figure 11 (b), MOSEV provides dNBR estimations of 287 and 327 for the low severity sites, which is relatively too large. In GFBS, the relative lower dNBR of 30 and 32 is provided at the corresponding low severity sites. The coarse resolution of MOSEV could also make it more difficult to capture the extreme values, as we found to be the case for the annual maximum forest fires in 2006 over CONUS. dNBR from GFBS clearly showed two peaks in the density plot of dNBR at around 100 and 700, representing the low and high severity, respectively. dNBR from MOSEV, however, showed only a single peak at around 500, indicating that the extreme low/high values in the 30m grid were averaged in the 500m grid. These findings reveal that burn severity from MOSEV has higher uncertainty for wildfires with larger spatial variabilities.

Another reason leading to the difference in the performances of the two data sets was related to sensors onboard Landsat and MODIS. MODIS has a wider spectral range and more spectral bands (36) than Landsat 7/8 (7 spectral bands/ 11 spectral bands, respectively), which resulted in different sensitivity to surface reflectance. For example, the NBR is commonly calculated using near-infrared (NIR) and shortwave infrared (SWIR) bands. In MOSEV, the bands used to calculate NBR are NIR: Band 2 (Range: 0.841–0.876 μm) and SWIR: Band 7 (Range: 2.105–2.155 μm). In GFBS, they are Landsat 5 Band 4 (Range: 0.76–0.90 μm) and SWIR: Band 7 (Range: 2.08–2.35 μm); Landsat 7 Band 4 (Range: 0.77–0.90 μm) and SWIR: Band 7 (Range: 2.09-2.35 μm); and Landsat 8 Band 5 (Range: 0.85–0.88 μm) and SWIR: Band 7 (Range: 2.11–2.29 μm). While MODIS and Landsat 8 are close in NIR and SWIR band information, Landsat 5 and 7 both have wider spectrums in NIR and SWIR than MODIS.

This study has shown that combining all available Landsat images, including those from Landsat 5, 7, and 8, could significantly improve the probability of obtaining dense cloud-free NBR time series. The NBR composite shows high spatial and temporal consistency with the NBR images closest to the start and end time of the fire event, despite different band settings used from Landsat 5, 7 and 8. Studies by Koutsias and Pleniou (2015) and Chen et al. (2020) also have shown that differences are small when using reflectance values from sensors aboard the Landsat 5, 7, and 8 satellites to calculate burn severity over burned area. While studies (Mallinis et al., 2018; Howe et al. 2022) have demonstrated that Sentinel-2 generally performed as well as Landsat 8 in burn severity mapping, the further extension of this study will also incorporate images from Sentinel-2 to obtain dNBR composite, especially on extending the GFBS data set to the present. With the finer spatial resolution (10 meter) and more frequent revisit period (5 days), GFBS could provide improved burn severity information when incorporating Sentinel-2 images. The National Aeronautics and Space Administration (NASA) has lounched the Harmonized Landsat and Sentinel-2 (HLS) project aiming to produce a seamless surface reflectance record from the Operational Land Imager (OLI) and Multi-Spectral

Instrument (MSI) aboard Landsat-8/9 and Sentinel-2A/B remote sensing satellites, respectively, which is an alternative source for extending the GFBS dataset (https://hls.gsfc.nasa.gov/)

With the development of radar-based techniques, Synthetic Aperture Radar (SAR) polarimetric images have been proven to be effective in burn severity mapping, owing to the strong correlation between SAR backscatter and burn severity [Czuchlewski and Weissel, 2005; Tanase et al., 2010; Tanase et al., 2011; Addisonand Oommen, 2018]. With the unique properties of L-band SAR, it is suitable for assessing and monitoring post-fire effects and burn severity [Tanase e al., 2010; Peacock et al., 2023]. For example, the frequency of L-band (1.26 GHz) allows it to penetrate through smoke, ash, and, to some extent, vegetation canopy. This capability makes L-band SAR particularly useful for assessing areas immediately after a fire, even in the presence of smoke or cloud cover that would obstruct optical sensors. The incorporation of L-band Synthetic Aperture Radar (SAR) data, such as the ALOS-2 PALSAR-2 ScanSAR Level 2.2 data (https://www.eorc.jaxa.jp/ALOS/en/alos-2/a2_about_e.htm) and and the incoming NASA-ISRO Synthetic Aperture Radar (NISAR,https://nisar.jpl.nasa.gov/), can also facilitate the retrieval of burn severity.

By comparing GFBS with CanLaBS, we found that the number of forest fires in CanLaBS dataset is larger than those in GFBS. This is because CanLaBS is based on the burn area map from Canada Landsat Disturbance product at 30 meter resolution, while GFBS is based on the burn area map from Global Fire Atlas which is derived from MODIS burn area product at 500 meter resolution. This difference in the spatial resolution of the burn area causes some small forest fires to be ignored in the GFBS dataset. Therefore, finer spatial resolution burn area product (10/30 meter) is promoted regionally and globally to better reveal the forest fire behavior, e.g. fire number, size and severity (Roy et al., 2019; Bar et al., 2020). Despite the differences in number of forest fires, GFBS agreed well to CanLaBS in terms of the annual forest burn severity. While the method to generate GFBS remains consistent, with the small difference to be ignored in banding settings from Landsat 5,7 and 8, GFBS provides comprehensive temporal coverage spanning from 2003 to 2016 for forest burn severity, indicating the potential application of GFBS in long term analysis of burn severity for forest fires beyond Canada, i.e. regions over the globe, e.g. CONUS, Australia, where GFBS has been demonstrated to perform well against ground truth. Moreover, integrating the 30 meter GFBS into the regional forest planning can enhance fire resilience in vulnerable areas, shaping policies that prioritize the forest environment [Bradley et al., 2016]. As climate change exacerbates the frequency, intensity, and unpredictability of wildfires globally, the analysis on GFBS data can help to assess the impact of these fires on carbon emissions [Xu et al., 2020], forest recovery [Meng et al., 2018], and biodiversity [Huerta et al., 2022], which would in turn inform predictive models that project future fire behavior under various climate scenarios.

**5. Conclusion**

We have introduced a newly developed dataset, named GFBS, which provides forest burn severity information with global coverage for the period 2003–2016. We identified global forest fires by overlaying the Global Fire Atlas data with the annual land cover data, MCD12Q1, and proposed an automated algorithm for calculating the severity of these fires. The algorithm used the band information from Landsat 5, 7, and 8 surface reflectance imagery to compute the most used burn severity spectral indices (dNBR and RdNBR) with a 30m spatial resolution and provide the output depicted in the GFBS dataset. Comparison between CanLaBS and GFBS showed good agreement in representing the

distribution of forest burn severity over Canada. The validation against field assessed burn severity category data in southeastern Australia showed that GFBS could provide burn severity estimation with clear differentiation between the high-severity class and moderate/low severity class of the in situ data, while such differences among burn severity class were not obvious in the MOSEV dataset. The validation results over CONUS showed dNBR values from GFBS to be more strongly correlated with CBI (r = 0.63) than dNBR from MOSEV (r = 0.28). RdNBR from GFBS also showed better agreement with CBI (r = 0.56) than RdNBR from MOSEV (r = 0.20). Thus, this database could be more reliable than prior sources of information for future studies of forest burn severity at global scale, as well as for studies to which forest burn severity could be relevant, such as in forest management and $CO_2$ emissions research.

A future direction for this study would be to extend the GFBS dataset to the present based on updated Global Fire Atlas data or other datasets providing global burn area and burn date information. Another direction is to involve more ground validations from the fire prone areas like south Africa and south Mexico to further evaluate and improve the performances of GFBS data globally.

**Competing interests**: The authors declare they have no conflict of interest.

**Data availability**: The GFBS data are freely accessible at https://doi.org/10.5281/zenodo.10037629 (He et al., 2023)

**Author contributions**: KH and EA designed and organized the manuscript. KH and XS prepared the related materials and ran the models for generating GFBS and the related assessments. XS and EA made contributions to the scientific framework of this study and discussed the interpretation of the results. All authors discussed the results and commented on the manuscript.

**Acknowledgments**: This research was supported by a National Science Foundation HDR award entitled "Collaborative Research: Near Term Forecast of Global Plant Distribution Community Structure and Ecosystem Function." Kang He received the support of the China Scholarship Council for four years' Ph.D. study at the University of Connecticut (under grant agreement no. 201906320068). Thanks for Rachael Gallagher and Eli Bendall from Western Sydney University for sharing the field assessed fire severity category data over southeastern Australia.

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
