# Peer review of "A Global Forest Burn Severity Dataset from Landsat Imagery"

_Earth System Science Data, 2023_

## Referee Comment (RC1)

The study developed a global 30-m resolution forest burn severity database. Unlike most previous studies that focused on fire occurrence, this data is about the severity of fire disturbance, which is evaluated by the amounts of biomass that were consumed by fire. Although the method is quite simple and not very innovative, it is an impressive work to produce 30 m resolution data products at the global scale between 2003 and 2016. That's why I think the data is valuable. However, some problems also exist.

Major comments:

1) Can you compare your product with the Canadian Landsat Burn Severity (CanLaBS) product which is also produced using 30 m resolution Landsat images? Does your method have any advantage over CanLaBS product?

2) Fig. 5 & Fig. 7: I think the number of data points in these figures is far from enough for you to conclude that your product is better than MOSEV. For example, in subfigure (b), there is only one low CBI value and one high CBI value. More data is needed for validation. Or, you can try other data sources such as statistical data to validate your data.

3) In the discussion part, we suggest you to discuss on whether the incorporation of L-band SAR data (e.g., LSAR-2 ScanSAR Level 2.2 product) can facilitate the retrieval of forest biomass before and after the fire.

4) Will the difference between band settings in Landsat-8 and Landsat5-7 induce temporal discontinuity in the burn severity product? Have you checked whether this product is suitable for long-term temporal analysis (e.g., the trend of burn severity). It's quite important information for users of this product.

5) You should stress the usefulness of this product, for example, by pointing out some potential applications.

Minor comments:

1) Line 96-98: These sentences should be removed, as Lines 99~106 are already the abstract of the method, which is simple enough for readers to understand.

2) Line 101: You should point out why NBR can reflect the biomass before or after the burn. References are also needed here.

3) Line 133: It would be better if you can explain why the denominator is the square root the NBR before burn, rather than the NBR before burn.

4) Line 136: I recommend you to mention how CBI is measured.

5) Line 144: I recommend to change the title into: forest fire coverage of Landsat composites.

6) Line 161-164: I recommend to move these sentences, including the figure, to section 2.4.

7) Line 169 and afterwards: Do you think it's better to move the comparisons with MOSEV to the Discussion part?

8) Lines 166-184: You should change your expression. As your product is not always and absolutely more correct, you should not say "MOSEV overestimate or underestimate ..." so confidently. You could add "might", for example.

9) I think Fig. 8 appears more convincing than Fig. 5 and Fig. 7. At least the amount of in-situ data is adequate.

10) Lines 238~245: What are the potential explanations to these differences?

11) Line 253: How can the spatial pattern comparison lead to the conclusion that your product has

improved accuracy than MOSEV? More explanations and proofs are needed.

12) Line 259~261: You should provide more proof, such as references, to conclude that MOSEV has truly over/under-estimated the burn severity in these regions, which your product has avoided.

13) You should pay attention to the writing. For example, in Line 298, "$CO_2$". Please check and correct such errors.

---

## Referee Comment (RC2)

A review of "A Global Forest Burn Severity Dataset from Landsat Imagery" by He et al. This manuscript attempts to develop a global forest burn severity (GFBS) database by combining Fire Atlas product, MODIS land cover data product, and the Landsat reflectance product. Results were validated over CONUS using the MODIS global burn severity dataset (MOSEV). Overall, the technical description seems technically sound and, in most cases, is well-written.  The experiment is designed for global, thus the influence of the dataset will be important. The results are reasonable, some issues need further clarification.

**General questions:**
Q1: Need further validations across the world, because the current version includes no validations outside CONUS. At least should done for Australia and North Russia where server burns have been reported.

Q2: What's the resolution of your data product is not clearly described in the abstract. It should be the strength of the developed GFBS database.

Q3: What's the spatial resolution of the CONUS-wide Composite Burn Index? If CBI plots are at 30m diameters, they should be more consistent with Landsat-derived burn servility.

Q4: When describing correlation, suggest using r, not R2.
e.g. "… dNBR of GFBS was more strongly correlated with CBI (R2 = 0.4) than dNBR of MOSEV (R2 = 0.08) …"

Q5: Is the method used in this study limited to the combustion areas already discovered by MODIS at 250m scale? If that's the case, it may miss many visible combustions on the Landsat scale (30m).

Figure 1: Why "Forest fire" in "I. Data input" was not linked to items in other sections?

**Specific comments:**
The definition of "burn severity" is not clear, e.g.,
L13: "… amounts of biomass"
L44: "… degree of biomass"

L49: Did you solve this issue?
"… the use of inadequate sampling data to construct the plot level prediction models"

L172: What's the reason for underestimation and overestimation?

L263: "We based GFBS on Landsat (5, 7, and 8) images …" please rewrite to read.

---

## Author Comment (AC1)

The study developed a global 30-m resolution forest burn severity database. Unlike most previous studies that focused on fire occurrence, this data is about the severity of fire disturbance, which is evaluated by the amounts of biomass that were consumed by fire. Although the method is quite simple and not very innovative, it is an impressive work to produce 30 m resolution data products at the global scale between 2003 and 2016. That's why I think the data is valuable. However, some problems also exist.

**Respond:** We appreciate the reviewer's constructive comments on the manuscript to further improve the quality and the contribution of our work. Below are the authors' responses on all of the reviewer's questions and suggestions. The reviewer's comments are marked as **red**, while our responses are marked as **blue**.

Major comments:

1) Can you compare your product with the Canadian Landsat Burn Severity (CanLaBS) product which is also produced using 30 m resolution Landsat images? Does your method have any advantage over CanLaBS product?

**Respond**: In the revised manuscript, we compared the fire severities of GFBS and MOSEV with the CanLaBS dataset for forest fires over Canada from 2003 to 2015. The results show that GFBS agreed well in representing the distribution of forest fire severity to those of CanLaBS over Canada, which represent a better agreement than the MOSEV dataset. In terms of the main advantages, both GFBS and CanLaBS are based on Landsat and the methods for deriving burn severity, e.g., dNBR, are similar, but GFBS dataset is global. As shown in Section 3, GFBS provides reasonable burn severity estimations not only in Canada, but also in fire prone areas globally, such as over CONUS and Australia, as validated in this study.

**From line 190 to line 194 in the revised manuscript:**

"In addition to validation against in-situ data., we also compared the fire severity magnitudes of GFBS with the CanLaBS dataset available over Canada. CanLaBS provides burn severity information for burned areas identified from the Canada Landsat Disturbance product at the level of individual 30m resolution pixels. The dataset was derived from Landsat imagery and uses values of pre-fire to post-fire differences in dNBRs for nearly 60 million hectares of burned areas across Canada's forests from 1985 to 2015. [Guindon et al., 2017; Guindon et al., 2018]."

References:

Guindon, L., P. Villemaire, R. St-Amant, P.Y. Bernier, A. Beaudoin, F. Caron, M.

Bonucelli and H. Dorion.: Canada Landsat Disturbance (CanLaD): a Canada-wide Landsat-based 30-m resolution product of fire and harvest detection and attribution since 1984. https://doi.org/10.23687/add1346b-f632-4eb9-a83d-a662b38655ad, 2017.

Guindon, L.; Bernier, P.Y.; Gauthier, S.; Stinson, G.; Villemaire, P.; Beaudoin, A.: Missing forest cover gains in boreal forests explained. Ecosphere, 9 (1), e02094. https://doi.org/10.1002/ecs2.2094, 2018.

**From line 233 to line 259 in the revised manuscript:**

3.2 Comparison between GFBS and CanLaBS over Canada

"In this section we describe the comparison of the fire severity maps of GFBS and MOSEV datasets to the ones from the CanLaBS dataset over Canada for an overlapped period from 2003 to 2015. Figure 6 shows the number and the trend of forest fires over Canada from 2003 to 2015, by CanLaBS data and GFBS products, while the vertical bar represents the number of forest fires recorded by both CanLaBS and GFBS each year. Due to the different sources and algorithms to map the burn area, the number of forest fires depicted by CanLaBS is larger than those by GFBS each year. Nevertheless, it is noted that GFBS agrees with CanLaBS in terms of the variations of forest fire activities, such as the intense forest fires in 2004 and 2015 and the relatively low number of forest fires in 2007 and 2008."

[Figure]

**Figure 6. Number of forest fires by CanLaBS and GFBS dataset. Vertical bars show the number of overlapping forest fires.**

"Figure 7 illustrate the scatterplots of dNBR of forest fires from CanLaBS against those from GFBS (panel a) and MOSEV (panel b), for the period 2003 to 2015. Consistent to the results shown in Figure 6, dNBR from GFBS shows strong correlation with the

dNBR from CanLaBS with r being 0.77 and a slightly underestimation of the overall dNBR for forest fires (bias = -12.42%). On the other hand, dNBR from MOSEV exhibited low correlation with the dNBR from CanLaBS (r = 0.42) and slight overestimation (bias = 11.84 %). Figure 7 (c) displays the probability density function (PDF) plots of CanLaBS dNBR, GFBS dNBR and MOSEV dNBR. It is noted the close PDFs of GFBS dNBR and CanLaBS dNBR, though the mode of GFBS distribution is at slightly lower dNBR value relative to the CanLaBS distribution. On the other hand, the distribution of MOSEV dNBR significantly deviates from CanLaBS dNBR, having a lower peak and larger tails."

[Figure]

(a)

(b)

(c)

**Figure 7. Scatterplots of dNBR from CanLaBS against those from (a) GFBS and (b) MOSEV; (c) density plot of dNBR from CanLaBS, GFBS and MOSEV, for forest fires from 2003 to 2015 over Canada.**

"Figure 8 presents the boxplots of distributions of dNBR from CanLaBS, GFBS and MOSEV separate by year. Consistent to the previous results, GFBS compares well with CanLaBS in terms of the dNBR distribution of annual forest fires and as well as the variations of dNBR over time, even though it provides slightly lower dNBR values compared to CanLaBS. On the other hand, MOSEV compared poorly with CanLaBS annual dNBR distributions, exhibiting overall larger dNBR values and larger anomalies over time."

[Figure]

**Figure 8. Boxplots of annual distributions of dNBR values from CanLaBS, GFBS and MOSEV for forest fires over Canada from 2003 to 2015.**

2) Fig. 5 & Fig. 7: I think the number of data points in these figures is far from enough for you to conclude that your product is better than MOSEV. For example, in subfigure (b), there is only one low CBI value and one high CBI value. More data is needed for validation. Or, you can try other data sources such as statistical data to validate your data.

**Respond**: The scatterplots in Figure 5 (Figure 13 in the revised manuscript) and Figure 7 (Figure 15 in the revised manuscript) in the original manuscript show the validations of CBI against dNBR and RdNBR for some specific events, indicating GFBS correlated better with CBI than MOSEV and thus provided more reasonable burn severity estimates for these fire events. Figure 16 in the revised paper displays the validation results of CBI against dNBR and RdNBR from GFBS and MOSEV involving all CBIs over CONUS (around 1315). The results also show that GFBS has stronger correlations with CBI than MOSEV.

**From line 386 to line 390 in the revised manuscript:**

[revised manuscript text omitted]

3) In the discussion part, we suggest you to discuss on whether the incorporation of L-band SAR data (e.g., LSAR-2 ScanSAR Level 2.2 product) can facilitate the retrieval

of forest biomass before and after the fire.

**Respond**: Thanks for this suggestion. We have added the sentences in the discussion section addressing the use of SAR in burn severity mapping.

**From line 474 to line 483 in the revised manuscript:**

"With the development of radar-based techniques, Synthetic Aperture Radar (SAR) polarimetric images have been proven to be effective in burn severity mapping, owing to the strong correlation between SAR backscatter and burn severity [Czuchlewski and Weissel, 2005; Tanase et al., 2010; Tanase et al., 2011; Addisonand Oommen, 2018]. With the unique properties of L-band SAR, it is suitable for assessing and monitoring post-fire effects and burn severity [Tanase e al., 2010; Peacock et al., 2023]. For example, the frequency of L-band (1.26 GHz) allows it to penetrate through smoke, ash, and, to some extent, vegetation canopy. This capability makes L-band SAR particularly useful for assessing areas immediately after a fire, even in the presence of smoke or cloud cover that would obstruct optical sensors. The incorporation of L-band Synthetic Aperture Radar (SAR) data, such as the ALOS-2 PALSAR-2 ScanSAR Level 2.2 data (https://www.eorc.jaxa.jp/ALOS/en/alos-2/a2_about_e.htm) and and the incoming NASA-ISRO Synthetic Aperture Radar (NISAR,https://nisar.jpl.nasa.gov/), can also facilitate the retrieval of burn severity."


**Respond**: This is a good point. The NBR composites use reflectance information from both Lansat 5,7 and 8 with different acquisition times, we presented the progress of how to process the NBR composites and compared the NBR obtained from Landsat 7, 8 with different acquisition times. The results demonstrate that the NBR composite has high spatial and temporal consistency with the NBR images closest to the start and end time of the fire event, despite different band settings used from Landsat 5, 7 and 8. Some studies have also shown that the differences are small when using reflectance values from sensors aboard the Landsat 5, 7, and 8 satellites to calculate burn severity.

**From line 214 to line 232 in the revised paper:**

"Figure 4 shows the data process for a single post-NBR Landsat composite for the fire event that ended on 17 September 2015 in north Washington. The first prior image for NBR calculation was on 20 September 2015 from Landsat 8 (as image 1). The cloud and shadows are removed in image 1 after applying the cloud/shadow mask. The next available image on 21 September 2015 from Landsat 7 (as image 2) was then used to fill those gaps in image 1 and obtain a new Landsat composite (phase 1). The third available image on 29 September 2015 from Landsat 8 (as image 3), image on 15 October 2015 if needed, was adopted sequentially to fill the un-scanned gap pixels in phase 1 and generate the final post NBR image for this event. The process for pre-NBR image calculation is the same but in a reversed time-order from the start time of the fire event."

[Figure]

**Figure 4. NBR image process for Landsat composite, for the fire event ended on 17 September 2015 in north Washington.**

"The scatterplot in Figure 5 (a) shows the NBR values of the overlapping pixels in image 1 and image 2, with the associated distributions of NBR for the fire event. It is noted that NBR values in images 1 and 2 show high correlation (r = 0.96), relatively low bias (-23.81%) and similar probability densities, even though they are derived from two different Landsat images (Landsat 8 and Landsat 7). The scatterplot in Figure 5 (b) shows the NBR values of overlapping pixels in image 1 and image 3, with the associated distribution of NBR for the fire event. Similarly, NBR values in image 1 and image 3 have high correlation (r = 0.96) and low bias (12.30 %) and similar probability densities, even though they are derived from different times (9 days apart). The results indicate that the cloud-free NBR composite mosaicking of all available Landsat images has reasonable accuracy with high spatial and temporal consistency."

[Figure]

(a)                                                                                   (b)

**Figure 5. Scatterplots of overlapped pixel values in (a) image 1 and image 2; (b) image 1 and image 3.**

In addition, in the discussion part, we also discuss the application of Sentinel-2 images in burn severity mapping in the future work.

**From line 460 to line 473 in the revised manuscript:**

"This study has shown that combining all available Landsat images, including those from Landsat 5, 7, and 8, could significantly improve the probability of obtaining dense cloud-free NBR time series. The NBR composite shows high spatial and temporal consistency with the NBR images closest to the start and end time of the fire event, despite different band settings used from Landsat 5, 7 and 8. Studies by Koutsias and Pleniou (2015) and Chen et al. (2020) also have shown that differences are small when using reflectance values from sensors aboard the Landsat 5, 7, and 8 satellites to calculate burn severity over burned area. While studies (Mallinis et al., 2018; Howe et al. 2022) have demonstrated that Sentinel-2 generally performed as well as Landsat 8 in burn severity mapping, the further extension of this study will also incorporate images from Sentinel-2 to obtain dNBR composite, especially on extending the GFBS data set to the present. With the finer spatial resolution (10 meter) and more frequent revisit period (5 days), GFBS could provide improved burn severity information when incorporating Sentinel-2 images. The National Aeronautics and Space Administration (NASA) has lounched the Harmonized Landsat and Sentinel-2 (HLS) project aiming to produce a seamless surface reflectance record from the Operational Land Imager (OLI) and Multi-Spectral Instrument (MSI) aboard Landsat-8/9 and Sentinel-2A/B remote sensing satellites, respectively, which is an alternative source for extending the GFBS dataset (https://hls.gsfc.nasa.gov/)"

**Respond**: The difference between GFBS and MOSEV burn severity mainly comes from the gap in spatial resolution. The coarse resolution of MOSEV (500 m) impede it to present the localized variability of burn severity while GFBS can with the improved resolution (30 m). We have presented two cases as Figure 11 (a) and (b) in the revised manuscript to demonstrate that MOSEV tends to provide burn severity estimation with large uncertainty.

**From line 285 to line 301 in the revised manuscript:**

"As mentioned above, MOSEV gave relatively small dNBR values in the event on October 15 2023, where burn severity is classified from in situ measurement as high. Figure 11 (a) displays the location of the ground verification sites with the corresponding burn severity class and associated dNBR values from MOSEV and GFBS. It is noted that within one MOSEV grid cell (500 meter) four ground verification sites are located. The dNBR value from MOSEV is 295 for all four sites, while three of the sites are classified as low and only one site is classified as high severity. On the other hand, at GFBS resolution (30 meter), we can note significant spatial variation in dNBR, with GFBS dNBR being 239 for the site classified as high and 9, 16 and 68 for the sites classified as low severity. In a surrounding MOSEV pixel we note a site classified as high severity, but dNBR from MOSEV is 188 while dNBR from GFBS is 397. In the event on October 21 2023, we found that MOSEV gave relatively high dNBR values at ground verification sites that are classified as low severity. Figure 11 (b) shows the locations of ground verification sites with corresponding classified burn severity and associated dNBR values from MOSEV and GFBS. In the two adjacent MOSEV grids, the dNBR values from

MOSEV are 287 and 327 respectively where both sites are classified as low severity. At GFBS resolution more significant changes between high and low dNBR are found within the same MOSEV grid, resulting in dNBR values of 30 and 32 for the ground verification sites classified as low severity. The results demonstrate the significance of GFBS high resolution data in representing the small-scale variations of dNBR and providing more granular and reliable dNBR estimations."

[Figure]

Figure 11. The location of ground verification sites with burn severity classes overlaid by dNBR values from GFBS and MOSEV for the fire event of (a) October 15 2023 and (b) October 21 2023.

11) Line 253: How can the spatial pattern comparison lead to the conclusion that your product has improved accuracy than MOSEV? More explanations and proofs are

needed.

**Respond**: In the revised manuscript, we additionally compared GFBS and MOSEV with CanLaBs product over Canada and validated GFBS and MOSEV using the field assessed burn severity data over southeastern Australia, and presented those results in two new sections (section 3.2 and section 3.3). The results show that GFBS performed better than MOSEV in terms of burn severity distribution and burn severity categorization. Two fire cases presented in Figure 11 (a) and (b) show the gap between resolution of GFBS and MOSEV can lead to significantly different burn severity estimates, where GFBS was shown to be more agreeable to ground truth.

12) Line 259~261: You should provide more proof, such as references, to conclude that MOSEV has truly over/under-estimated the burn severity in these regions, which your product has avoided.

**Respond**: In the revised manuscript, we provided the comparison results between GFBS, MOSEV and CanLaBS over Canada, indicating a better agreement of GFBS to CanLaBS in terms of dNBR. Besides, we also provided the validation results of ground verified burn severity data against dNBR from GFBS and MOSEV over southeastern Australia, demonstrating a better ability of GFBS to differentiate the burn severity between different categories. A detailed comparison between GFBS and MOSEV at some field assessed burn severity sites shown that, MOSEV tends to overestimate the dNBR at the low severity and underestimate the dNBR at high severity site, while GFBS could provide more reasonable dNBR estimates at the field assessed site.

13) You should pay attention to the writing. For example, in Line 298, "CO2". Please check and correct such errors.

**Respond**: We have corrected those errors in the revised manuscript.

---

## Author Comment (AC2)

This manuscript attempts to develop a global forest burn severity (GFBS) database by combining Fire Atlas product, MODIS land cover data product, and the Landsat reflectance product. Results were validated over CONUS using the MODIS global burn severity dataset (MOSEV). Overall, the technical description seems technically sound and, in most cases, is well-written. The experiment is designed for global, thus the influence of the dataset will be important. The results are reasonable, some issues need further clarification.

**Respond**: We appreciate the reviewer's constructive comments on the manuscript to further improve the quality and the contribution of our work. Below are the authors' responses to all of the reviewer's questions and suggestions. The reviewer's comments are marked as **red**, while our responses are marked as **blue**.

 **General questions:**

Q1: Need further validations across the world, because the current version includes no validations outside CONUS. At least should done for Australia and North Russia where server burns have been reported.

**Respond**: In the revised manuscript, we compared the performances of GFBS with CanLaBS data for forest fires over Canada from 2003 to 2015. The results show that GFBS agreed well in representing the distribution of forest fire severity to those of CanLaBS over Canada, which outperforms MOSEV.

**From line 190 to line 194 in the revised manuscript:**

[revised manuscript text omitted]

Q2: What's the resolution of your data product is not clearly described in the abstract. It should be the strength of the developed GFBS database.

**Respond:** Thanks for your suggestions, we have addressed this in the revised abstract

**From line 12 to line 14 in the revised manuscript:**

"To improve quantification of the intensity and extent of forest fire damage, we have developed a 30-meter resolution Global Forest Burn Severity (GFBS) dataset of the degree of biomass consumed by fires from 2003 to 2016."

Q3: What's the spatial resolution of the CONUS-wide Composite Burn Index? If CBI plots are at 30m diameters, they should be more consistent with Landsat-derived burn servility.

**Respond:** Yes. The CBI plots for CONUS is with a 30-m diameter, which is described in https://burnseverity.cr.usgs.gov/products/cbi. How CBI is measured in also mentioned in the revised manuscript:

**from line 161 to line 166:**

"CBI was developed by Key and Benson (2006) to assess the aboveground effects of fire on vegetation and soil land use types (i.e., burn severity). It is determined through direct field observations after a fire when assessors visited various sites within the burned area to evaluate the effects of the fire on different components of the ecosystem, such as the degree of charring, percentage of foliage consumed, changes in ground cover, and mortality of plants. The CBI score for each site was calculated by averaging the scores of the different components. This overall score represents the burn severity at a specific site."

Q4: When describing correlation, suggest using r, not R2.
e.g. "… dNBR of GFBS was more strongly correlated with CBI (R2 = 0.4) than dNBR of MOSEV (R2 = 0.08) …"
**Respond:** Thanks for your suggestion. We have used r value to replace $R^2$ value in the revised manuscript, and also use r value in the figures.

For example, from line 22 to line 26:
"Using the CONUS-wide Composite Burn Index (CBI) as a ground truth, we showed that dNBR from GFBS was more strongly correlated with CBI (r = 0.63) than dNBR from MOSEV (r = 0.28). RdNBR from GFBS also exhibited better agreement with CBI (r = 0.56) than RdNBR from MOSEV (r = 0.20)."

Q5: Is the method used in this study limited to the combustion areas already discovered by MODIS at 250m scale? If that's the case, it may miss many visible combustions on the Landsat

scale (30m).

**Respond:** Yes, you are correct. The fire polygon data we used to estimate forest burn severity is from the Fire Atlas product derived from the MODIS burn area product at 500 m resolution. fires with burn area smaller than 500 m might not be detected. By comparing GFBS with CanLaBS over Canada, we found that the number of forest fires in CanLaBS dataset is larger than those in GFBS. This is because CanLaBS retrieves burn area from Canada Landsat Disturbance product at 30 meter resolution. This difference in the spatial resolution of burn area causes some small forest fires unidentified in the GFBS dataset.

We have addressed this in the discussion section,

**From line 484 to line 490:**

"By comparing GFBS with CanLaBS, we found that the number of forest fires in CanLaBS dataset is larger than those in GFBS. This is because CanLaBS is based on the burn area map from Canada Landsat Disturbance product at 30 meter resolution, while GFBS is based on the burn area map from Global Fire Atlas which is derived from MODIS burn area product at 500 meter resolution. This difference in the spatial resolution of the burn area causes some small forest fires to be ignored in the GFBS dataset. Therefore, finer spatial resolution burn area product (10/30 meter) is promoted regionally and globally to better reveal the forest fire behavior, e.g. fire number, size and severity (Roy et al., 2019; Bar et al., 2020)."

Figure 1: Why "Forest fire" in "I. Data input" was not linked to items in other sections?

**Respond:** We apologize for the mistake, the forest fire is firstly determined by overlaying the Fire Atlas polygon with land cover map, and then the corresponding start and end dates for the forest fires are used for dNBR calculation. We have designed the flowchart to make it more clear to describe the steps.

[Figure]

**Figure 1. Methodology for building the GFBS database (2003–2016) and validation and comparison with the MOSEV benchmark.**

**Specific comments:**

The definition of "burn severity" is not clear, e.g.,

L13: "... amounts of biomass"

L44: "... degree of biomass"

**Respond:** Thanks for pointing out this, the burn severity is derived for determining the degree of biomass consumption and the overall impact of fire on ecosystems, as mentioned in Keeley, 2009.

We have corrected this.

**From line 12 to line 13 in the revised manuscript,**

"To improve quantification of the intensity and extent of forest fire damage, we have developed a 30-meter resolution Global Forest Burn Severity (GFBS) dataset of the degree of biomass consumed by fires from 2003 to 2016."

L49: Did you solve this issue?

"... the use of inadequate sampling data to construct the plot level prediction models"

**Respond:** We are sorry, but we didn't find this statement in the original manuscript in line 49

and throughout the original submitted manuscript.

**In line 49 in the original manuscript:**

"Regionally, the Monitoring Trends in Burn Severity (MTBS) dataset, which includes burn severity assessments for the contiguous United States (CONUS) and provides information on fire perimeters and severity classes, uses satellite data—specifically, Landsat imagery (Eidenshink et al., 2007)."

L172: What's the reason for underestimation and overestimation?

**Respond:** The main reason for the differences in dNBR estimates between GFBS and MOSEV lays the gaps between the resolution of MOSEV (500 meter) and GFBS (30 meter). The coarse resolution of MOSEV impedes it to capture the localized variability of dNBR. In the revised paper, we have presented two cases as Figure 11 (a) and (b) to demonstrate that MOSEV tends to provide burn severity estimation with large uncertainty.

**From line 285 to line 301 in the revised manuscript:**

"As mentioned above, MOSEV gave relatively small dNBR values in the event on October 15 2023, where burn severity is classified from in situ measurement as high. Figure 11 (a) displays the location of the ground verification sites with the corresponding burn severity class and associated dNBR values from MOSEV and GFBS. It is noted that within one MOSEV grid cell (500 meter) four ground verification sites are located. The dNBR value from MOSEV is 295 for all four sites, while three of the sites are classified as low and only one site is classified as high severity. On the other hand, at GFBS resolution (30 meter), we can note significant spatial variation in dNBR, with GFBS dNBR being 239 for the site classified as high and 9, 16 and 68 for the sites classified as low severity. In a surrounding MOSEV pixel we note a site classified as high severity, but dNBR from MOSEV is 188 while dNBR from GFBS is 397. In the event on October 21 2023, we found that MOSEV gave relatively high dNBR values at ground verification sites that are classified as low severity. Figure 11 (b) shows the locations of ground verification sites with corresponding classified burn severity and associated dNBR values from MOSEV and GFBS. In the two adjacent MOSEV grids, the dNBR values from MOSEV are 287 and 327 respectively where both sites are classified as low severity. At GFBS resolution more significant changes between high and low dNBR are found within the same MOSEV grid, resulting in dNBR values of 30 and 32 for the ground verification sites classified as low severity. The results demonstrate the significance of GFBS high resolution data in representing the small-scale variations of dNBR and providing more granular and reliable dNBR estimations."

[Figure]

**Figure 11. The location of ground verification sites with burn severity classes overlaid by dNBR values from GFBS and MOSEV for the fire event of (a) October 15 2023 and (b) October 21 2023.**

**And from line 435 to line 443 in the revised manuscript:**

"As shown in Figure 11 (a), stemming from the coarse spatial resolution, MOSEV provides dNBR value of 295 for the site classified as high severity as well as for those classified as low severity, leading to an overestimation for low severity sites. With the improved spatial resolution, GFBS is able to capture more detailed localized variability of dNBR, providing more reasonable dNBR estimation for low severity sites (dNBR equal to 9, 16, 68). Similarly, in the event shown in Figure 11 (b), MOSEV provides dNBR estimations of 287 and 327 for

the low severity sites, which is relatively too large. In GFBS, the relative lower dNBR of 30 and 32 is provided at the corresponding low severity sites."

L263: "We based GFBS on Landsat (5, 7, and 8) images …" please rewrite to read.

**Respond:** We have corrected these kinds of writing errors in the revised manuscript.

---

## Author Response (AR2)

We appreciate the reviewer's suggestions on the manuscript to further improve the quality and the contribution of our work. Below are the authors' responses to all of the reviewer's suggestions. The reviewer's suggestions are marked as **red**, while our responses are marked as **blue**.

Suggestions for revision or reasons for rejection
(visible to the public if the article is accepted and published)
This study's general summary and significance should be reaffirmed at the end of the abstract.

**Respond**: We have summarized and addressed the contributions of our dataset at the end of the abstract.

From line 31 to line 33 in the revised manuscript:

"The GFBS dataset provides a more precise and reliable assessment of burn severity than existing available datasets. These enhancements are crucial for understanding the ecological impacts of forest fires and for informing management and recovery efforts in affected regions worldwide."

Fig2. The reason for underestimation and overestimation should be further clarified.

**Respond:** Just to clarify, figure 2 only shows the locations of ground verification burn severity sites over southeastern Australia and forest fire CBIs over CONUS. The figure does not contain any results.

[Figure]

(a)          (b)

**Figure 2. Locations of (a) ground verification burn severity sites over southeastern Australia and (b) forest fire CBIs over CONUS.**

If the reviewer is referring to the comparison between the GFBS and MOSEV against ground validation, we have discussed the differences from line 285 to line 301 in the revised manuscript:

"As mentioned above, MOSEV gave relatively small dNBR values in the event on October 15 2023, where burn severity is classified from in situ measurement as high. Figure 11 (a) displays the location of the ground verification sites with the corresponding burn severity class and associated dNBR values from MOSEV and GFBS. It is noted that within one MOSEV grid cell (500 meter) four ground verification sites are located. The dNBR value from MOSEV is 295 for all four sites, while three of the sites are classified as low and only one site is classified as

high severity. On the other hand, at GFBS resolution (30 meter), we can note significant spatial variation in dNBR, with GFBS dNBR being 239 for the site classified as high and 9, 16 and 68 for the sites classified as low severity. In a surrounding MOSEV pixel we note a site classified as high severity, but dNBR from MOSEV is 188 while dNBR from GFBS is 397. In the event on October 21 2023, we found that MOSEV gave relatively high dNBR values at ground verification sites that are classified as low severity. Figure 11 (b) shows the locations of ground verification sites with corresponding classified burn severity and associated dNBR values from MOSEV and GFBS. In the two adjacent MOSEV grids, the dNBR values from MOSEV are 287 and 327 respectively where both sites are classified as low severity. At GFBS resolution more significant changes between high and low dNBR are found within the same MOSEV grid, resulting in dNBR values of 30 and 32 for the ground verification sites classified as low severity. The results demonstrate the significance of GFBS high resolution data in representing the small-scale variations of dNBR and providing more granular and reliable dNBR estimations."

Fig9. The image seems a little blur.
**Respond:** We have provided a high-resolution image in the revised manuscript.

[Figure]

**Figure 9. Spatial patterns of dNBR for wildfires on (a) October 15 2023, (b) October 17 2023 and (c) October 21 2023, in southeastern Australia, derived from the GFBS and MOSEV datasets.**